# A novel biosensor to study cAMP dynamics in cilia and flagella

Shatanik Mukherjee[1†], Vera Jansen[2†], Jan F Jikeli[2†], Hussein Hamzeh[2], Luis Alvarez[1], Marco Dombrowski[2], Melanie Balbach[1], Timo Strünker[1,3], Reinhard Seifert[1], U Benjamin Kaupp[1]*, Dagmar Wachten[2]*

[1]Department of Molecular Sensory Systems, Center of Advanced European Studies and Research, Bonn, Germany; [2]Minerva Max Planck Research Group, Molecular Physiology, Center of Advanced European Studies and Research, Bonn, Germany; [3]Centrum für Reproduktionsmedizin und Andrologie, Universitätsklinikum Münster, Münster, Germany

*For correspondence: U.B. Kaupp@caesar.de (UBK); dagmar.wachten@caesar.de (DW)

†These authors contributed equally to this work

Competing interests: The authors declare that no competing interests exist.

**Abstract** The cellular messenger cAMP regulates multiple cellular functions, including signaling in cilia and flagella. The cAMP dynamics in these subcellular compartments are ill-defined. We introduce a novel FRET-based cAMP biosensor with nanomolar sensitivity that is out of reach for other sensors. To measure cAMP dynamics in the sperm flagellum, we generated transgenic mice and reveal that the hitherto methods determining total cAMP levels do not reflect changes in free cAMP levels. Moreover, cAMP dynamics in the midpiece and principal piece of the flagellum are distinctively different. The sole cAMP source in the flagellum is the soluble adenylate cyclase (SACY). Although bicarbonate-dependent SACY activity requires $Ca^{2+}$, basal SACY activity is suppressed by $Ca^{2+}$. Finally, we also applied the sensor to primary cilia. Our new cAMP biosensor features unique characteristics that allow gaining new insights into cAMP signaling and unravel the molecular mechanisms underlying ciliary function *in vitro* and *in vivo*.

## Introduction

Cyclic adenosine 3′,5′-monophosphate (cAMP) controls various physiological functions, such as the heart beat (*Zaccolo, 2009*), learning and memory (*Lee, 2015*; *Morozov et al., 2003*), olfaction (*Kaupp, 2010*), and fertilization (*Buffone et al., 2014*). cAMP is synthesized by adenylate cyclases (ACs) and degraded by phosphodiesterases (PDEs) (*Francis et al., 2011*; *Hanoune et al., 1997*; *Steegborn, 2014*). Local cAMP signaling is achieved by targeting of signaling components to sub-cellular compartments and assembly of signaling complexes (*Willoughby and Cooper, 2007*). A prime example of a subcellular compartment is the cilium, where cAMP translates external stimuli into cellular responses (*Johnson and Leroux, 2010*).

Cilia come in two different flavors: primary cilia and motile cilia. A prominent example of a motile cilium is the sperm flagellum that serves as a sensory antenna and propels sperm forward. cAMP-signaling pathways control different functions during the sperm's journey to the egg. The principal cAMP source in sperm, the bicarbonate- and $Ca^{2+}$-sensitive soluble adenylate cyclase SACY (*Esposito et al., 2004*; *Hess et al., 2005*; *Vacquier et al., 2014*; *Xie et al., 2006*), is activated by increasing bicarbonate concentrations (*Luconi et al., 2005*; *Wennemuth et al., 2003*) after sperm are released from the epididymis. Sperm lacking SACY are immotile, which causes male infertility (*Esposito et al., 2004*; *Hess et al., 2005*; *Xie et al., 2006*). The elevated cAMP levels activate protein kinase A (PKA) - the primary cAMP target in sperm (*Nolan et al., 2004*). PKA activation in turn accelerates the flagellar beat (*Hess et al., 2005*; *Wennemuth et al., 2003*; *Xie et al., 2006*). cAMP is also involved in a maturation process of sperm called capacitation that is essential for fertilizing

**eLife digest** Cells can change the way they grow, move or develop in response to information from their environment. This information is first detected at the surface of the cell and then the information is relayed around the interior of the cell by signaling molecules known as "second messengers". A molecule called cAMP is a well-known second messenger that is involved in many different signaling pathways. Therefore, the levels of cAMP in specific areas of the cell need to be precisely regulated to enable different signaling pathways to be activated at specific times and locations.

Some cells have hair-like structures called cilia or flagella on their surface. Cilia and flagella are able to move the fluid that surrounds the cells or even move the cells themselves. The second messenger cAMP plays an essential role in making cilia move, but it is challenging to analyze the dynamics of cAMP – that this, how the levels of this molecule change over time – in these structures. The levels of cAMP in live cells can only be measured using fluorescent biosensors. Introducing these biosensors into specific cell structures is difficult and they are not sensitive enough to respond to low levels of cAMP. Furthermore, it is difficult to measure cAMP activity inside such tiny structures using these biosensors.

Mukherjee, Jansen, Jikeli et al. now address some of these challenges by creating a new cAMP biosensor that has several unique features. Most importantly, it can respond to very low levels of cAMP, making it more sensitive than previous biosensors. Mukherjee et al. test this new biosensor in the flagella of sperm cells from mice, which reveals how the production of cAMP is regulated in the flagellum. The new biosensor also shows that different parts of the flagellum can have different cAMP dynamics. In the future, this new biosensor could be used to study cAMP in other structures and compartments within cells.

the egg; stimulation of cAMP synthesis by bicarbonate seems to be required for protein tyrosine phosphorylation, which is a hallmark of capacitated sperm (*Visconti et al., 1995*). However, the molecular mechanisms underlying these cAMP-signaling pathways in sperm are not well understood.

Components of cAMP signaling have also been identified in immotile cilia, e.g. in the specialized cilium of mammalian olfactory neurons, where cAMP stimulates the electrical signal evoked by odorants (*Kaupp, 2010*). Primary cilia also host several cAMP-signaling components (*Johnson and Leroux, 2010*): the somatostatin 3 receptor (SSTR3), various adenylate cyclases (AC3, AC4, AC6, AC8), PKA, and Epac2 (*exchange protein directly activated by cAMP*) (*Bishop et al., 2007*; *Händel et al., 1999*; *Kwon et al., 2010*; *Masyuk et al., 2006*; *Ou et al., 2009*). An important signaling pathway in primary cilia is controlled by Sonic hedgehog (Shh), which determines neuronal cell fate during development (*Chiang et al., 1996*; *Ericson et al., 1997*; *Huangfu et al., 2003*). Shh signaling relies on PKA activation (*Mukhopadhyay et al., 2013*; *Tuson et al., 2011*). However, the physiological function of cAMP signaling in primary cilia and the underlying molecular mechanisms are ill-defined.

Analyzing cAMP dynamics *in vivo* became feasible by the advent of genetically-encoded cAMP biosensors that rely on FRET (*Förster resonance energy transfer*) (*Castro et al., 2014*; *Hong et al., 2011*; *Sprenger and Nikolaev, 2013*; *Willoughby and Cooper, 2008*). The most advanced cAMP sensors are based on the cyclic nucleotide-binding domain (CNBD) of Epac1/2 (*Nikolaev et al., 2004*; *Ponsioen et al., 2004*) or HCN channels (*Nikolaev et al., 2006*). The CNBD is sandwiched between the fluorescent donor CFP (cyan fluorescent protein) and the fluorescent acceptor YFP (yellow fluorescent protein). The $K_D$ values of the Epac1/2- or HCN2-based cAMP sensors range between 1–15 µM (*Hong et al., 2011*; *Sprenger and Nikolaev, 2013*; *Willoughby and Cooper, 2008*), suitable to detect basal cAMP concentrations and changes in cAMP levels in the micromolar range (*Börner et al., 2011*). In some cell systems, these sensors were able to report a cAMP increase, but their sensitivity was not sufficient to determine basal cAMP levels (*Börner et al., 2011*).

The analysis of cAMP dynamics in cilia and flagella is challenging for several reasons. First, the cAMP biosensor must be targeted to this cellular compartment; second, measuring low cAMP concentrations requires the affinity of the sensor to be high, in particular, in a subcellular compartment

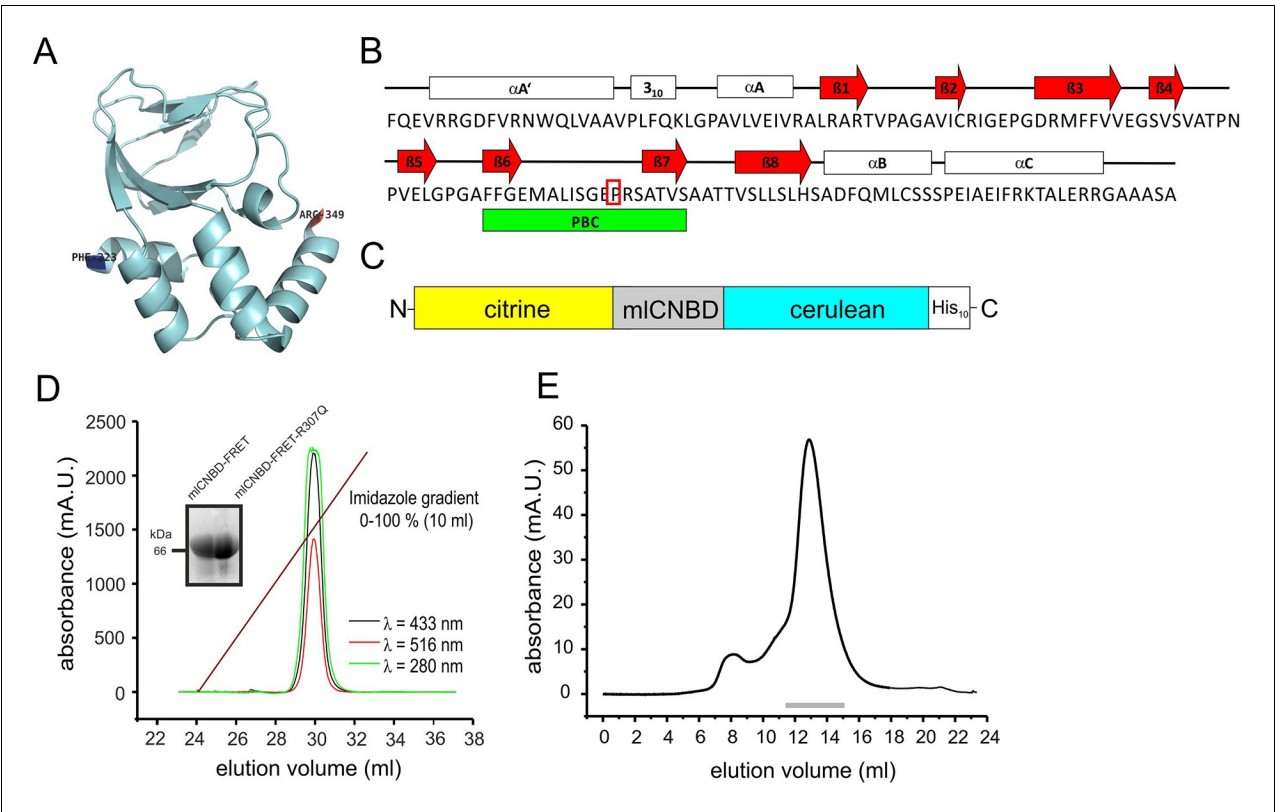

**Figure 1.** Generation and purification of mlCNBD-FRET. (**A**) Ribbon presentation of a single mlCNBD according to (**Schünke et al., 2011**). The first (Phe223) and the last amino acid (Arg349) are indicated. (**B**) Structural features of mlCNBD. Alpha helices (αA-C), beta rolls (β1–8), and the phosphate binding-cassette (PBC) are indicated. The arginine (R) crucial for cAMP-binding is boxed. (**C**) The mlCNBD-FRET biosensor. The sensor has been generated by fusing citrine and cerulean to the N- and C-terminus of mlCNBD, respectively. For purification, a $His_{10}$ tag has been added to the C-terminus. (**D**) Purification of mlCNBD-FRET via cobalt immobilized-metal affinity chromatography. Representative elution profile for mlCNBD-FRET using a linear imidazole gradient. The absorption has been recorded at three different wavelengths (280 nm: protein, green; 433 nm: cerulean, black; 516 nm: citrine, red). The inset shows a representative Western blot for the purified mlCNBD-FRET and mlCNBD-FRET-R307Q protein, stained with an anti-His antibody. (**E**) Size-exclusion chromatography of the purified mlCNBD-FRET protein. Representative elution profile. The protein eluted in a main peak at 67 kDa (peak maximum), close to the expected molecular mass of 70.9 kDa. A minor peak was observed that eluted earlier and represents the void volume. Fractions indicated by the grey line have been used for analysis.

with only a few cAMP molecules present. Finally, quantitative characterization and calibration in cilia is technically demanding due to their small size compared to the cell soma.

$Ca^{2+}$dynamics have been measured in primary cilia using genetically-encoded biosensors. These studies revealed that the cilium represents a unique $Ca^{2+}$-signaling compartment that is functionally distinct from the cytoplasm (**DeCaen et al., 2013**; **Delling et al., 2013**). Whether cAMP signaling in cilia and flagella is also functionally distinct from the cytoplasm, is ill-defined.

Here, we describe a new FRET-based cAMP biosensor (mlCNBD-FRET) that is built from the CNBD of the bacterial *Mloti*K1 channel (**Nimigean et al., 2004**) and binds cAMP with nanomolar affinity (**Cukkemane et al., 2007**; **Peuker et al., 2013**). This biosensor features unique characteristics that enable its application in solution, in cell lines, and *in vivo* using kinetic, fluorometric, and live-cell imaging techniques. To target the sensor to cilia and flagella, we designed a cilia-specific targeting approach, and we generated transgenic mice expressing mlCNBD-FRET exclusively in the sperm flagellum. Our results reveal that the dynamics of total and free cAMP levels in sperm and the cAMP dynamics in the midpiece and principal piece of the flagellum are distinctively different. We investigated the regulation of cAMP dynamics in sperm and obtained new insights into the $Ca^{2+}$-dependent control of cAMP synthesis.

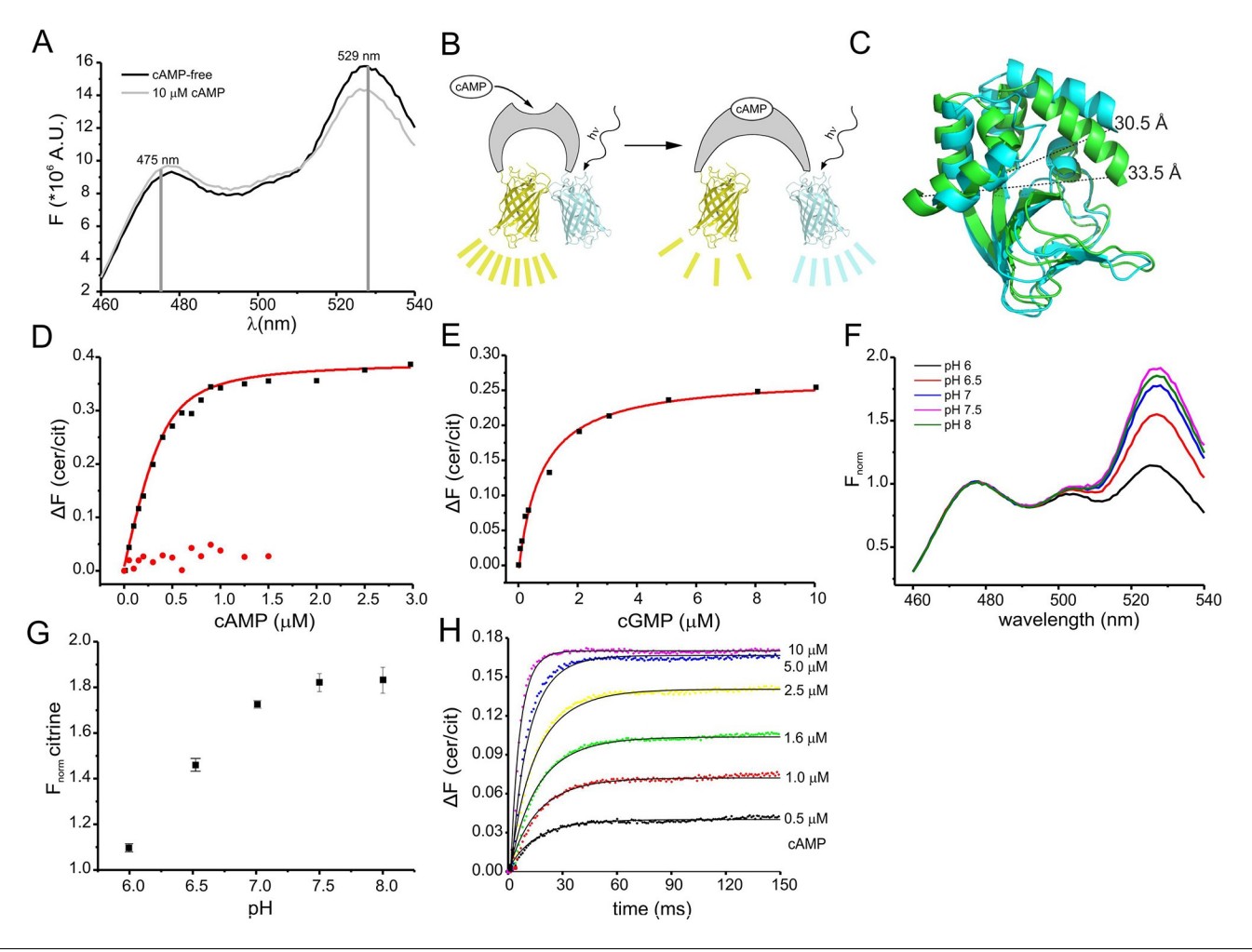

**Figure 2.** Characterization of the purified mlCNBD-FRET. (**A**) Fluorescence spectra of mlCNBD-FRET at 430 nm excitation before (black) and after addition of 10 µM cAMP (grey). (**B**) Schematic representation of the structural changes evoked by cAMP upon binding, FRET becomes smaller, indicating that cerulean and citrine move further apart. (**C**) Structural changes occurring after cAMP binding. The cAMP-free structure is shown in blue, the cAMP-bound structure is shown in green. Distances are presented in Ångstrom. (**D**) Binding of cAMP to mlCNBD-FRET (black) determined by fluorescence spectroscopy. Representative experiments showing an increase in the baseline-corrected cerulean/citrine emission ratio (△F) of mlCNBD-FRET (430 nm excitation) after cAMP binding. Data have been fitted using a single binding-site model (red line) (*Cukkemane et al., 2007*). As a control, mlCNBD-FRET-R307Q (red dots) has been used. Measurements have been performed using 1 µM protein. (**E**) Representative experiments showing an increase of △F of mlCNBD-FRET (430 nm excitation) after cGMP binding. Data has been fitted (see (**D**), red line). Measurements have been performed using 1 µM protein. (**F**) Normalized fluorescence spectra of mlCNBD-FRET (430 nm excitation) at different pH conditions. Spectra were normalized to the cerulean emission at 471 nm. (**G**) Normalized FRET (430 nm excitation, 529 nm emission) at different pH values. Data have been taken from measurements shown in (**F**) and are presented as mean ± S.D.; n = 3. (**H**) Kinetics of cAMP binding to mlCNBD-FRET measured using the stopped-flow technique. Different cAMP concentrations (in µM: 0.5, 1, 1.6, 2.5, 5, and 10) were mixed with the purified mlCNBD-FRET protein (2.5 µM) and the change in FRET was measured over time. Solid lines represent a global fit of a one-step model (see materials and methods) with the following parameters: $k_{on}$ = 2.6 $*10^7$ $M^{-1}s^{-1}$ and $k_{off}$ = 12.8 $s^{-1}$.

## Results

### Characterization of the mlCNBD-FRET protein

The CNBD from the bacterial *Mloti*K1 channel (mlCNBD) consists of a prototypical eight-stranded beta roll (β1–8) flanked by three alpha helices (αA-C, *Figure 1A,B*) (*Clayton et al., 2004*; *Cukkemane et al., 2007*; *Nimigean et al., 2004*). A phosphate-binding cassette (PBC) that interacts with the sugar and phosphate moiety of the cyclic nucleotide is the most conserved feature of

CNBDs (*Figure 1B*). Upon cAMP binding, mlCNBD undergoes a conformational change (*Schünke et al., 2011*). We fused two variants of the green fluorescent protein - citrine (*Griesbeck et al., 2001*) and cerulean (*Rizzo et al., 2004*) - to the N- and C-terminus of mlCNBD to measure cAMP-induced conformational changes by FRET. Cerulean, the FRET donor, transfers energy to the FRET acceptor citrine. A histidine tag (His$_{10}$) was fused to the C-terminus of cerulean (*Figure 1C*) to purify the protein via cobalt immobilized-metal affinity chromatography (*Figure 1D*). In size-exclusion chromatography, the mlCNBD-FRET protein eluted as a single peak with an apparent molecular mass of 67 kDa, close to the calculated molecular mass of 70.9 kDa (*Figure 1E*).

Binding of cAMP to mlCNBD-FRET was measured in a spectrofluorometer. The two fluophores in the protein undergo FRET, indicating that they are in close proximity in the absence of cAMP (*Figure 2A*). Addition of cAMP decreased FRET: the cerulean emission at 475 nm increased, whereas the citrine emission at 529 nm decreased (*Figure 2A*). Thus, upon cAMP binding, the two fluorophores move further apart (*Figure 2B,C*). In the following, changes of the cerulean/citrine emission ratio ($\triangle F$) are used; an increase of this ratio reflects a cAMP increase and vice versa. We characterized cAMP binding to mlCNBD-FRET by measuring the change in $\triangle F$ caused by increasing cAMP concentrations (*Figure 2D*). The dose-response relationship was fitted with a single binding-site model (*Cukkemane et al., 2007*). The $K_D$ of mlCNBD-FRET for cAMP was $66 \pm 15$ nM (n = 5), similar to that for mlCNBD ($68 \pm 9$ nM) (*Cukkemane et al., 2007*). This demonstrates that the two fluorophores do not interfere with cAMP binding, and that the sensor detects cAMP concentrations in the low to medium nanomolar range. Finally, a control mutant carrying an R307Q amino-acid exchange in the PBC that abolishes cAMP binding (*Figure 1B*) (*Bubis et al., 1988*; *Harzheim et al., 2008*; *McKay and Steitz, 1981*; *Zagotta et al., 2003*), did not respond to cAMP concentrations up to 1.5 µM (*Figure 2D*).

We also determined the cGMP sensitivity of mlCNBD-FRET: cGMP enhanced $\triangle F$ at a tenfold higher concentration than cAMP (*Figure 2E*). From the dose-response relationship, we determined for cGMP a $K_D$ of $504 \pm 137$ nM (n = 6), which is similar to the $K_D$ value of mlCNBD ($499 \pm 69$ nM) (*Cukkemane et al., 2007*).

The fluorescence of GFP derivatives, in particular YFP and citrine, is pH sensitive (*Griesbeck et al., 2001*). Therefore, we measured the emission spectrum of mlCNBD-FRET (while exciting cerulean) at different pH values. The citrine emission peak at 529 nm increased at higher pH values and saturated at pH $\geq$ 7.5, whereas the cerulean emission at 475 nm was largely pH-insensitive (*Figure 2F,G*). The $K_D$ for cAMP binding was only modestly affected by changes in pH (pH 6: $101 \pm 5$ nM, pH 7: $98 \pm 21$ nM; pH 7.5: $66 \pm 15$ nM, pH 8: $50 \pm 13$; n = 3), indicating that the cAMP affinity might increase upon alkalization. Of note, it has been reported for other CNBDs that the cAMP affinity depends on the pH (*Gordon et al., 1996*; *Kaupp and Seifert, 2002*).

We also measured FRET using Fluorescence Lifetime Spectroscopy (FLS). The fluorescence lifetime of the donor decreases during FRET. Because cAMP binding to mlCNBD-FRET reduces FRET, the cerulean lifetime should increase upon binding. The decay of cerulean fluorescence alone was fitted with a bi-exponential function (weighted mean of the two lifetime constants $\tau_{wm} = 2.52 \pm 0.09$ ns, n = 12). In mlCNBD-FRET, the cerulean lifetime decreased ($\tau_{wm} = 2.38 \pm 0.04$ ns, n = 11). Saturating cAMP (5 µM) decreased FRET and, thereby, the cerulean lifetime increased ($\tau_{wm} = 2.44 \pm 0.03$ ns, n = 5). In summary, the mlCNBD-FRET sensor is suitable for both intensity- and lifetime-based approaches.

## Response kinetics of the FRET sensor

The response time of the sensor to rapid changes in cAMP was determined using the stopped-flow technique. Different cAMP concentrations were mixed with purified mlCNBD-FRET protein. From the time course of the FRET response, we determined the time constant for the changes in fluorescence by numerically fitting a simple one-step model to the data (*Figure 2H*, see materials & methods). According to this model, the overall *on* and *off* rates of ligand binding and changes in FRET are $2.5 \pm 0.6 * 10^7$ M$^{-1}$ s$^{-1}$ and $9.3 \pm 6.7$ s$^{-1}$ (n = 3). The *off* rate is rate-limiting, resulting in a time constant of about 100 ms. Thus, mlCNBD-FRET allows measuring cAMP changes on a 100 millisecond time scale.

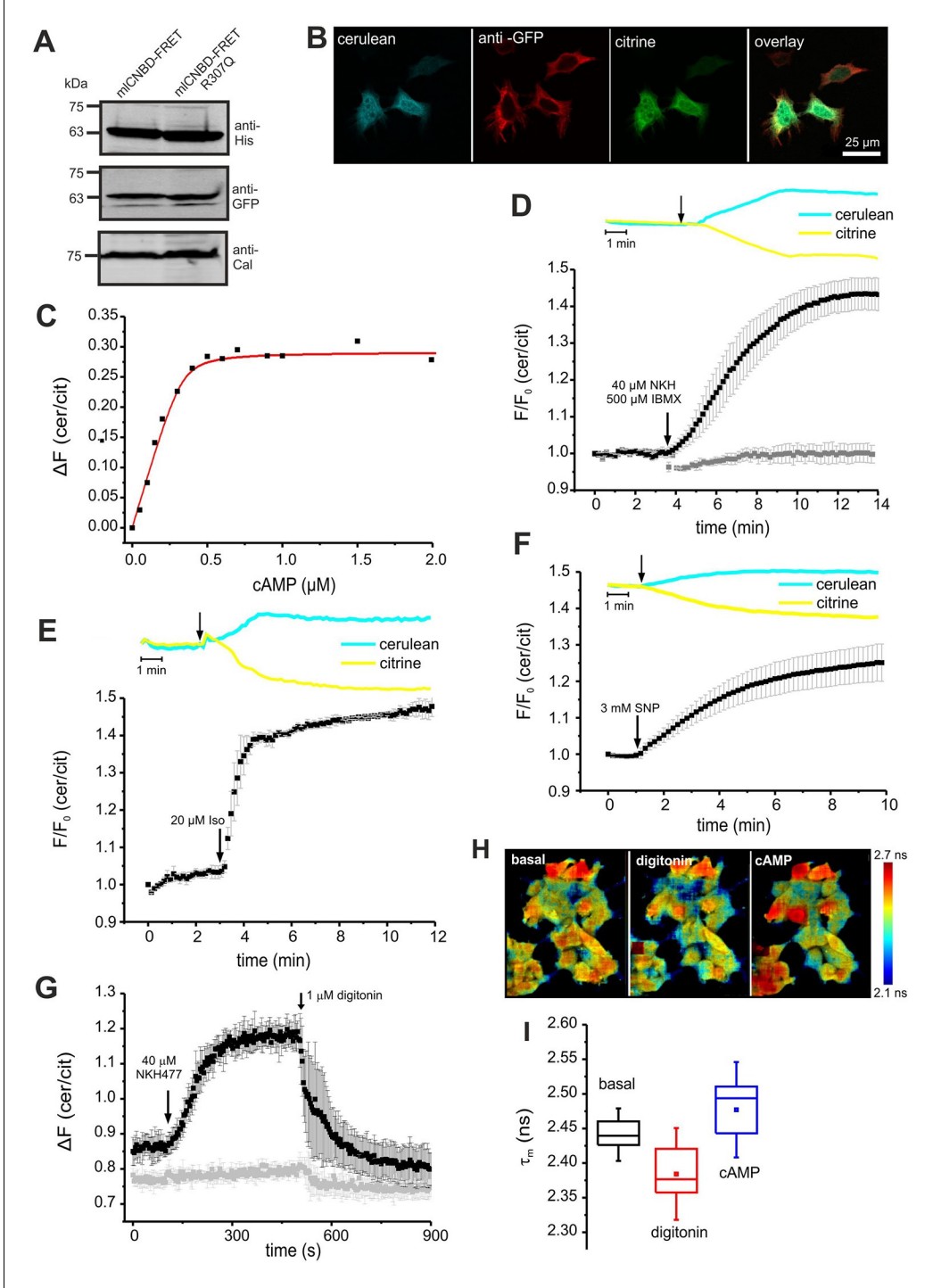

**Figure 3.** Characterization of mlCNBD-FRET in HEK293 cells. (**A**). Representative Western blot using total protein lysates from mlCNBD-FRET and mlCNBD-FRET-R307Q-expressing cells, stained with an anti-His and an anti-GFP antibody. Calnexin (Cal) has been used as a loading control. (**B**) Immunocytochemistry. HEK293 cells expressing mlCNBD-FRET (cerulean: blue, citrine: green) have been labeled with an anti-GFP antibody and a fluorescent secondary antibody (red). Scale bar 25 µm. (**C**) Ligand binding of cAMP to mlCNBD-FRET in HEK293 cells determined by fluorescence spectroscopy. Representative experiment showing an increase of the baseline-corrected cerulean/citrine emission ratio ΔF of mlCNBD-FRET (430 nm excitation) at different cAMP concentrations. Cells have been permeabilized with 20 µM digitonin before addition of cAMP. Data have been fitted using a single binding-site model (red line) (**Cukkemane et al., 2007**). (**D**) Changes in FRET in HEK293 cells expressing mlCNBD-FRET after stimulation with 40 µM NKH477/500 µM IBMX. FRET has been measured by fluorescence microscopy. Representative traces of raw data are shown above. HEK293 cells expressing mlCNBD-FRET-R307Q (grey) have been used as a control. Data are presented as mean ± S.D. (mlCNBD-FRET: n = 31; mlCNBD-FRETR307Q: *Figure 3 continued on next page*

*Figure 3 continued*

n = 3). (**E**) Changes in FRET in HEK293 cells expressing mlCNBD-FRET after stimulation with 2 µM isoproterenol. Representative traces for the raw data are shown above. Data are presented as mean ± S.D.; n = 9. (**F**) Changes in FRET in HEK293 cells expressing mlCNBD-FRET after stimulation with 3 mM SNP. Representative traces for the raw data are shown above. Data are presented as mean ± S.D.; n = 36. (**G**) Changes in FRET in HEK293 cells expressing mlCNBD-FRET (black) or mlCNBD-FRET-R307Q (grey) after stimulation with 40 µM NKH477. After reaching a steady-state, cells have been permeabilized using 1 µM digitonin. FRET has been measured using spectrofluorometer. Data are presented as mean ± S.D.; n = 3. (**H**) Changes in cerulean fluorescence lifetime using FLIM. HEK293 cells expressing mlCNBD-FRET were imaged under basal conditions, after addition of 20 µM digitonin, and the following addition of 5 µM cAMP. The cerulean fluorescence decay was recorded and fitted with a bi-exponential decay to calculate the lifetime. Data are presented as mean ± S.D. Representative images are shown using a look-up table ranging from 2.1 ns (blue) to 2.7 ns (red). (**I**) Mean values of the two lifetimes averaged over different regions of interest in part (**H**); n = 8 for each condition.

The following figure supplement is available for figure 3:

**Figure supplement 1.** Characterisation of mlCNBD-FRET in HEK293 cells.

## Characterization of mlCNBD-FRET in mammalian cell lines

We tested mlCNBD-FRET in a cellular environment using HEK293 cells. Western blotting confirmed the expression of mlCNBD-FRET (*Figure 3A*); the sensor was mainly localized to the cytosol, but a fraction also resided in the nucleus (*Figure 3B*).

To determine the cAMP-binding characteristics, we calibrated mlCNBD-FRET by bathing digitonin-permeabilized HEK293 cells in defined cAMP concentrations. The $K_D$ for cAMP binding was 73 ± 20 nM (n = 11, *Figure 3C*), which is similar to that of the purified mlCNBD-FRET (66 ± 15 nM, *Figure 2D*). Using the $K_D$ value and the cAMP null-point calibration, we determined a basal free cAMP concentration of 35 ± 1 nM (n = 3, *Figure 3—figure supplement 1A*).

Next, we studied the cAMP dynamics of mlCNBD-FRET-expressing HEK293 cells by stimulation with a mixture of 40 µM NKH477 and 500 µM IBMX: NKH477 activates transmembrane adenylate cyclases that synthesize cAMP (tmAC) (*Hosono et al., 1992*), whereas IBMX inhibits phosphodiesterases (PDEs) that hydrolyze cAMP (*Schmidt et al., 2000*). NKH477/IBMX treatment increased △F, reflecting an increase of cAMP levels (*Figure 3D*). Of note, the changes in citrine and cerulean emission are of opposite sign but similar time course, demonstrating that the changes in fluorescence were owing to FRET and not to fluorescence artefacts. The FRET change commenced within a few seconds after stimulation and reached a steady-state after 10 min (ratio increase: 43 ± 4%, n = 31, *Figure 3D*). HEK293 control cells expressing the cAMP-insensitive mlCNBD-FRET-R307Q mutant did not respond (n = 3, *Figure 3D*). We also studied whether the mlCNBD-FRET sensor reports a cAMP increase mediated by G protein-coupled receptors: stimulation with 20 µM isoproterenol, an agonist of beta-adrenergic receptors, rapidly changed △F by 47 ± 3% (n = 9, *Figure 3E*). mlCNBD-FRET also reports changes in cGMP; stimulation with 3 mM SNP, which releases nitric oxide (NO) that activates soluble guanylyl cyclases (*Denninger and Marletta, 1999*), changed △F by 25 ± 5% (n = 36, *Figure 3F*). Similar results were obtained using cell populations: NKH477/IBMX and isoproterenol treatment both evoked a larger change than SNP (n = 3, *Figure 3—figure supplement 1B–D*). The twofold difference in the maximal changes evoked by drugs that stimulate cAMP- or cGMP-synthesis factors seems small, considering that the respective $K_D$ values differ by 10fold. Probably, at rest, the sensor is partially occupied by cAMP and stimulation with NKH477/IBMX or isoproterenol saturates the response. To test the sensor with submaximal agonist concentrations, we stimulated cells with increasing concentrations of NKH477 and analyzed the dose-response relationship (*Figure 3—figure supplement 1F*). The $EC_{50}$ for NKH477 was 3.6 ± 0.6 µM (n = 4).

Moreover, we analyzed whether mlCNBD-FRET also reliably reports a decrease of cAMP levels. HEK293 cell populations were pre-stimulated with NKH477 until steady-state before permeabilizing with 1 µM digitonin to release cAMP (*Figure 3G*). In turn, the FRET ratio decreased (*Figure 3G*), demonstrating that mlCNBD-FRET registers both, an increase and decrease of the intracellular cAMP concentration. To rule out any unspecific effects during permeabilization, we used mlCNBD-FRET-R307Q-expressing cells as a control. Here, the FRET ratio remained constant after NKH477 stimulation; only a small decrease occurred upon addition of digitonin (*Figure 3G*).

Similarly, we tested the reversibility of the sensor by alternatingly stimulating with isoproterenol followed by a wash-out step. Stimulation with 500 nM isoproterenol increased the FRET ratio

(*Figure 3—figure supplement 1G*). When reaching a maximum, the stimulus was removed and in turn, the FRET ratio decreased. Afterwards, a second stimulus of 500 nM isoproterenol was applied, which resulted in a similar increase as observed for the first stimulus (*Figure 3—figure supplement 1G*).

Finally, we also tested the performance of mlCNBD-FRET in a cellular environment by two different lifetime-based techniques. First, using FLS, we calibrated the sensor with defined cAMP concentrations in permeabilized, mlCNBD-FRET-expressing HEK293 cells. The cerulean lifetime increased after cAMP addition (*Figure 3—figure supplement 1E*). The $K_D$ for cAMP derived from the dose-lifetime relationship was $99.0 \pm 10.1$ nM (n = 3), which is similar to the $K_D$ derived from fluorescence intensity-based FRET ($73 \pm 20$ nM). Furthermore, the basal free cAMP concentration in HEK293 cells, calculated from lifetime and intensity-based measurements, was similar ($48.7 \pm 11.0$ nM vs. $35 \pm 1$ nM, respectively, n = 3). To evoke cAMP changes in intact cells, cells were treated with NKH477/IBMX, which also prolonged the cerulean lifetime compared to controls ($\tau_{wm} = 1.88 \pm 0.04$ ns vs. $\tau_{wm} = 1.98 \pm 0.03$ ns, n = 9).

Second, we used FRET Fluorescence Lifetime Imaging (FLIM). In mlCNBD-FRET-expressing HEK293 cells, the cerulean lifetime decreased as a result of FRET ($\tau_{wm} = 2.44 \pm 0.02$ ns vs. $\tau_{wm} = 3.28 \pm 0.07$ ns for cerulean only, n = 9). Upon addition of NKH477/IBMX, the cerulean lifetime increased when FRET decreased ($\tau_{wm} = 2.50 \pm 0.02$ ns, n = 9). To explore the dynamic range, we first permeabilized cells to release intracellular cAMP, followed by addition of a saturating cAMP concentration (5 µM). The shift of lifetime upon changes in cAMP is illustrated in the color-coded FLIM images (*Figure 3H*). The cerulean lifetime decreased after digitonin treatment from $\tau_{wm} = 2.44 \pm 0.02$ ns to $\tau_{wm} = 2.38 \pm 0.04$ ns (n = 8), reflecting a cAMP decrease. Upon addition of cAMP, the lifetime increased to $\tau_{wm} = 2.47 \pm 0.05$ ns (n = 8, *Figure 3H,I*). In summary, mlCNBD-FRET is an exquisitely sensitive biosensor for measuring cAMP dynamics in the nanomolar range, preferably using fluorescence intensity techniques, but also using lifetime-based techniques.

## Expression and analysis of mlCNBD-FRET in mouse sperm

To study cAMP dynamics in sperm flagella, we generated transgenic mice expressing mlCNBD-FRET under the control of the protamine 1 promoter (*Prm1*, *Figure 4A*). Transgenic mice were generated by pronuclear injection using standard procedures (*Ittner and Götz, 2007*). Genomic insertion of the transgene was confirmed by PCR (*Figure 4B*). The mlCNBD-FRET protein was exclusively expressed in sperm and was mainly targeted to the flagellum (*Figure 4C–E*). *Prm1*-mlCNBD-FRET males were fertile, demonstrating that mlCNBD-FRET does not impair sperm function (*Figure 4—source data 1*).

To calibrate mlCNBD-FRET in sperm, we followed a similar strategy as used for HEK293 cells. Titration of digitonin-treated sperm with cAMP decreased FRET, thereby increasing the FRET ratio (*Figure 5A*). The mean $K_D$ value derived from the dose-response relation was $103 \pm 31$ nM (n = 7, *Figure 5A*). We also measured by FLS the cerulean lifetime of mlCNBD-FRET sperm. At rest, the cerulean lifetime was $\tau_{wm} = 2.0 \pm 0.1$ ns (n = 4); addition of cAMP to digitonin-treated sperm prolonged the cerulean lifetime. At saturating cAMP concentrations (6 µM), the lifetime was $\tau_{wm} = 2.3 \pm 0.03$ ns (n = 3). The mlCNBD-FRET characteristics for cAMP binding in the different systems are summarized in a table (*Figure 5—source data 1*). For comparison, we have also included a summary of cAMP biosensors and their characteristics (*Figure 5—source data 2*). In summary, the sensor allows measuring changes in cAMP levels in sperm using fluorescence intensity and lifetime-based approaches.

## Dynamics of free and total cAMP in sperm

Changes in cAMP levels in sperm have previously been inferred from total cAMP concentrations measured with immunoassays. However, total cyclic nucleotide concentrations in cells are several orders of magnitude larger than free concentrations. Moreover, large volume ratios between the soma and small compartments like flagella might obscure the extent and time course of changes in cAMP. Therefore, we compared the dynamics of total and free cAMP in sperm upon stimulation with bicarbonate. Bicarbonate (25 mM) increased the FRET ratio, reflecting an increase of the free cAMP concentration (*Figure 5B*). cAMP commenced to rise within seconds, reached a steady-state after about 5 min, and persisted during bicarbonate stimulation (*Figure 5C*). In contrast, total cAMP

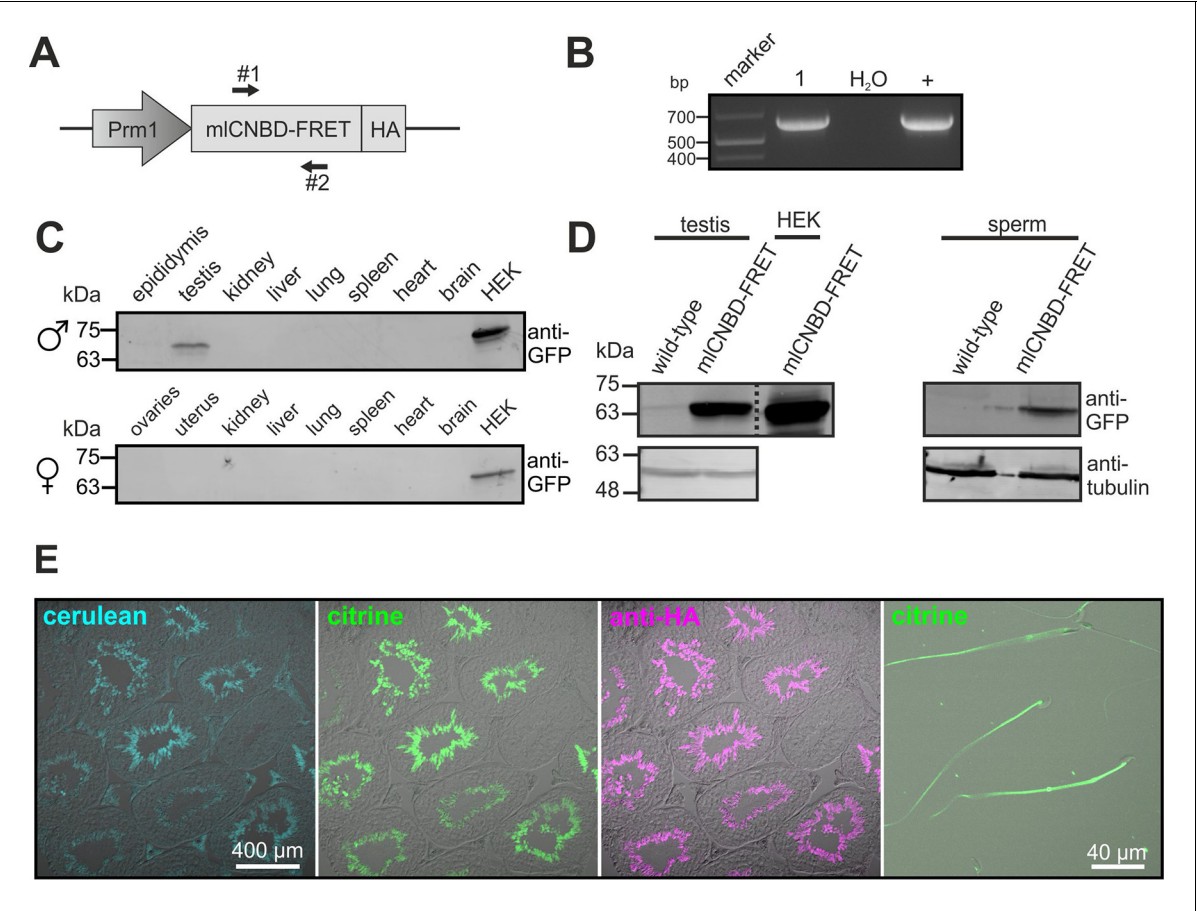

**Figure 4.** Generation of a *Prm1*-mlCNBD-FRET transgenic mouse line. (**A**) Scheme of the *Prm1*-mlCNBD-FRET targeting vector. Expression of hemagglutinin(HA)-tagged mlCNBD-FRET is driven by the Protamine 1 promoter (*Prm1*); arrows indicate the position of genotyping primers (#1, 2). (**B**) Genotyping by PCR. In *Prm1*-mlCNBD-FRET mice, a 653 bp fragment is amplified (1). The targeting vector served as a positive control (+). (**C**) Western blot analysis of mlCNBD-FRET expression in lysates from different tissues from a transgenic male and a female. Lysates from HEK293 cells expressing mlCNBD-FRET served as positive control. Proteins have been labeled using an anti-GFP antibody. (**D**) Western blot analysis of mlCNBD-FRET expression in testis and sperm lysates from a wild-type and a transgenic male. Lysates from HEK cells expressing mlCNBD-FRET served as positive control. Proteins have been labeled using an anti-GFP antibody; tubulin has been used as a loading control. (**E**) Immunohistochemical analysis of mlCNBD-FRET expression in testis and sperm. Cryosections of mouse testis were probed with anti-HA antibody and fluorescent secondary antibody (purple); the fluorescence for cerulean (blue) or citrine (green) is also shown.

The following source data is available for figure 4:

**Source data 1.** Fertility parameters of mlCNBD-FRET transgenic males.

levels, after a rapid rise (1 min), declined again to basal levels within 10 min (*Figure 5D*). Previous studies also reported a transient increase of cAMP during bicarbonate stimulation, although the extent and speed of recovery varied among species (*Battistone et al., 2013*; *Brenker et al., 2012*). In essence, the changes in total cAMP concentration unsatisfactorily reflect the changes in free cAMP levels in mouse sperm.

Of note, we tested whether the bicarbonate-induced increase in FRET ratio reflects a change in cAMP levels and not a change in the intracellular $pH_i$. Wild-type sperm were loaded with the fluorescent pH indicator BCECF and changes in $pH_i$ were measured after stimulation with 25 mM bicarbonate or 10 mM $NH_4Cl$ as a control. Stimulation with $NH_4Cl$, evoked a pronounced alkalization and, in turn, decreased the FRET ratio (*Figure 5—figure supplement 1A,B*). The kinetics of both changes were similar, indicating that the change in FRET is evoked by the change in $pH_i$. In contrast, stimulation with $NaHCO_3$ did not dramatically change $pH_i$ (*Figure 5—figure supplement 1A*). These results

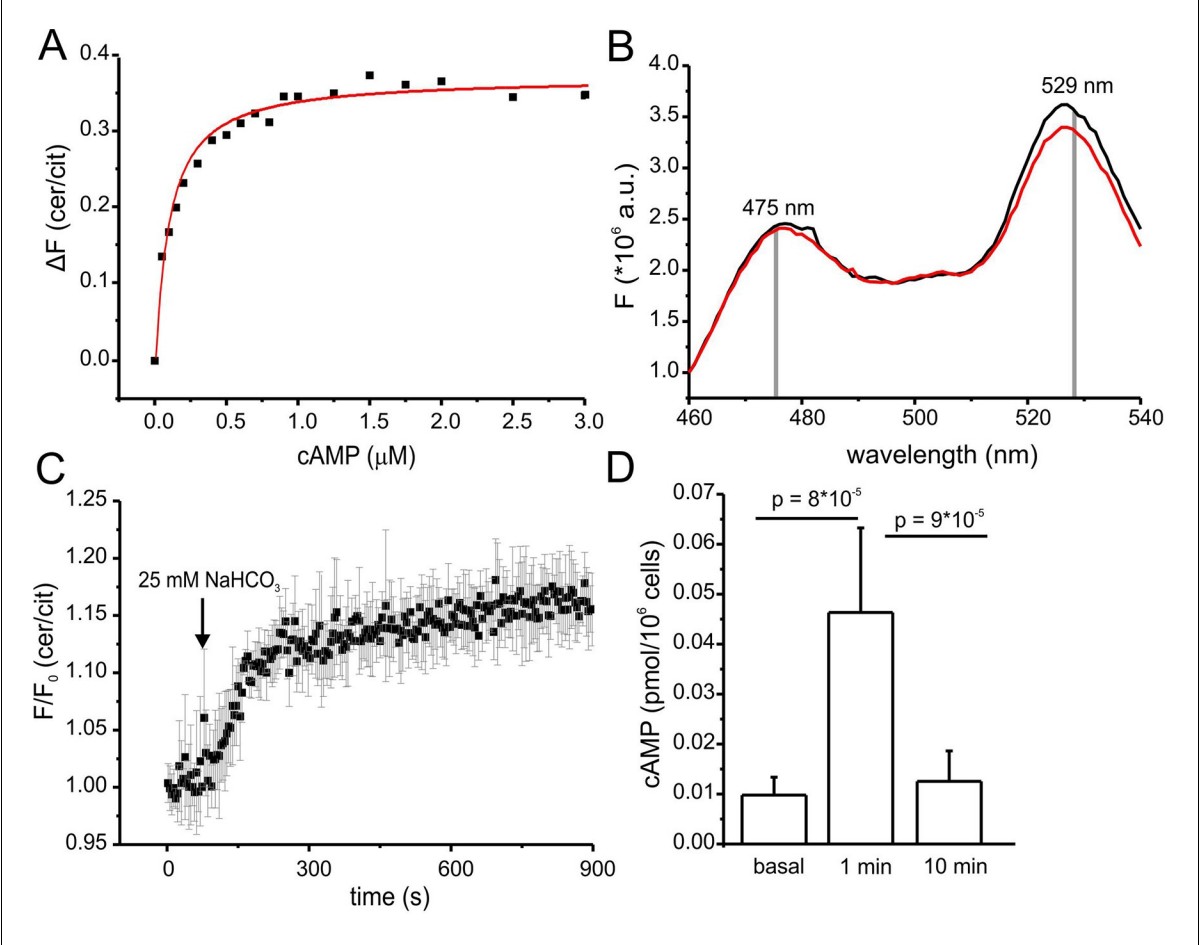

**Figure 5.** Characterization of cAMP dynamics in sperm. (**A**) Ligand binding of cAMP to mlCNBD-FRET in mouse sperm determined by fluorescence spectroscopy. Representative experiment showing an increase in the baseline-corrected cerulean/citrine emission ratio ($\triangle$F) of mlCNBD-FRET (430 nm excitation) after cAMP binding. Cells have been permeabilized with digitonin before addition of cAMP. Data have been fitted using a single binding-site model (red line) (*Cukkemane et al., 2007*); n = 7. (**B**) Fluorescence spectra of mlCNBD-FRET at 430 nm excitation before (black) and after stimulation for 5 min with 25 mM bicarbonate (red). (**C**) Changes in FRET after stimulation of a mlCNBD-FRET sperm with 25 mM bicarbonate. Sperm have been kept in 2 mM $Ca^{2+}$ buffer. FRET has been measured using a spectrofluorometer. Data is shown as mean ± S.D.; n = 3. (**D**) Total cAMP content. Sperm have been stimulated with 25 mM bicarbonate for 1 or 10 min and the total cAMP content has been determined using an immunoassay. The p values according to Students t-test are indicated. Data are shown as mean ± S.D.; n = 4.

The following source data and figure supplement are available for figure 5:

**Source data 1.** Characteristics of mlCNBD-FRET.
**Source data 2.** Characteristics of other cAMP biosensors.
**Figure supplement 1.** Characterisation of mlCNBD-FRET in mouse sperm.

support the notion that the changes in FRET evoked by $NaHCO_3$ reflect changes in cAMP rather than $pH_i$.

Expression of mlCNBD-FRET also allows spatially resolving cAMP dynamics in sperm. The mlCNBD-FRET biosensor is predominantly expressed in the flagellum (*Figure 4E*). Although the major if not only cAMP source is SACY (*Brenker et al., 2012*; *Esposito et al., 2004*; *Hess et al., 2005*), a debate continues about the presence and function of transmembrane adenylate cyclases (tmACs) (*Buffone et al., 2014*; *Vacquier et al., 2014*; *Wertheimer et al., 2013*), for a comprehensive discussion see (*Brenker et al., 2012*). To gain more insight, a spatio-temporal analysis of tmAC- and SACY-dependent changes in cAMP is required. Stimulation by NKH477 and bicarbonate

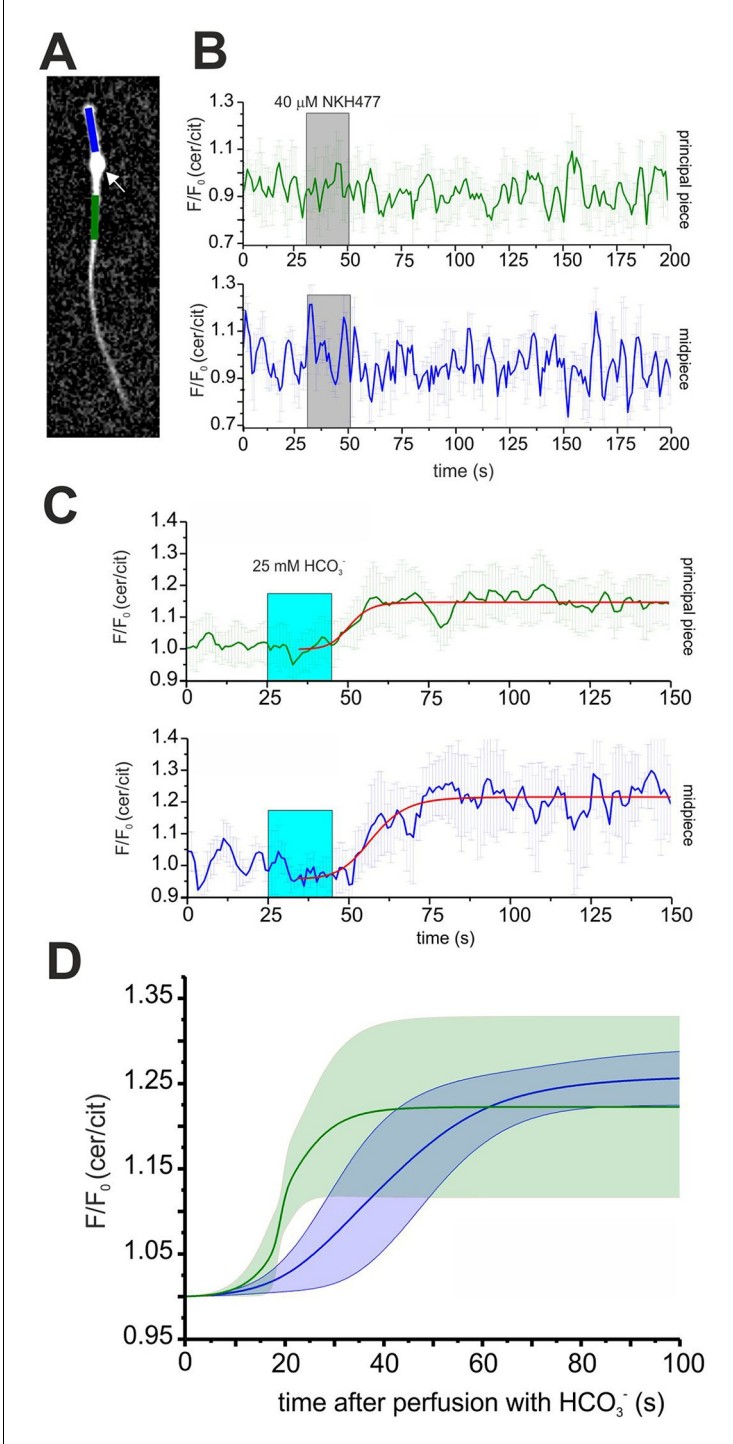

**Figure 6.** Spatio-temporal cAMP dynamics in the sperm flagellum. (A) cAMP dynamics was analyzed in a region 20 µm in length in the midpiece (blue) and principal piece (green) of freely beating sperm. The cytoplasmic droplet is indicated with an arrow. (B) Changes in FRET after stimulation with 40 µM NKH477. The perfusion with NKH477 is indicated with a grey box. Data for a representative cell are shown as mean ± S.D. in the midpiece (blue) and principal piece (green). (C) Changes in FRET after stimulation with 25 mM bicarbonate. The perfusion with bicarbonate is indicated with a blue box. Individual traces have been fitted using logistic regression (Origin 9) (red line). (D) Average of the fitted data presented in (C). The blue and green line represent the mean value for the midpiece and principal piece, respectively (n = 7). The blue and green areas represent the corresponding S.D.

discriminates between tmACs and SACY activity, respectively. Thus, we measured cAMP dynamics in the freely beating flagellum of mlCNBD-FRET sperm upon perfusion with 40 µM NKH477 or 25 mM bicarbonate. mlCNBD-FRET is equally distributed along the flagellum, which allows distinguishing between the cAMP dynamics in the midpiece and principal piece (*Figure 6A*; blue and green region, respectively). Invariably, no cAMP change in either the midpiece or principal piece was detected after perfusion with NKH477 ($0 \pm 3\%$, n = 3, *Figure 6B*). Thus, our results rule out the presence of tmACs in the sperm flagellum. However, perfusion with bicarbonate increased cAMP levels in both, midpiece ($26 \pm 3\%$ ) and principal piece ($22 \pm 10\%$ ) (n = 7, *Figure 6C*). To compare the response kinetics in the two compartments, individual traces were fitted using a logistic regression (Origin 9.0) and average data were plotted (*Figure 6D*). The kinetics of the cAMP increase by bicarbonate was faster in the principal piece than in the midpiece, indicating that cAMP dynamics in these two compartments is differently regulated.

## Regulation of cAMP dynamics by $Ca^{2+}$

SACY activity is controlled by bicarbonate and $Ca^{2+}$ (*Carlson et al., 2007*; *Chen et al., 2000*; *Jaiswal and Conti, 2003*; *Litvin et al., 2003*; *Wennemuth et al., 2003*). $Ca^{2+}$ enhances the *in vitro* activity of native or heterologously expressed SACY in a dose-dependent manner (*Jaiswal and Conti, 2003*). In intact sperm, several read-outs including total cAMP content, flagellar beat frequency, stimulation of PKA activity, and increase of tyrosine phosphorylation have been used to assess SACY activity (*Carlson et al., 2007*; *Navarrete et al., 2015*; *Visconti et al., 1995*; *Wennemuth et al., 2003*). For example, stimulation of sperm with bicarbonate in so-called nominally $Ca^{2+}$-free buffers, which contain an undefined free $Ca^{2+}$ concentration $[Ca^{2+}]_o$ in the low micromolar range (*Marín-Briggiler et al., 2005*), does not increase the flagellar beat frequency (*Carlson et al., 2007*; *Wennemuth et al., 2003*). Recently, it has been proposed that $Ca^{2+}$ regulates cAMP signaling in a biphasic manner (*Navarrete et al., 2015*), i.e. cAMP signaling seems to be stimulated at extremely low $[Ca^{2+}]_o$. Although $[Ca^{2+}]_i$ is not as well defined as $[Ca^{2+}]$ in cell-free assays (*Jaiswal and Conti, 2003*), it is reasonable to assume that $[Ca^{2+}]_o$ affects $[Ca^{2+}]_i$ accordingly. Given the limitations of indirect read-outs of free cAMP concentrations, we studied the $Ca^{2+}$ regulation of SACY using mlCNBD-FRET.

We followed the changes in cAMP upon stimulation with 25 mM bicarbonate in buffer with defined $Ca^{2+}$ concentrations of 10 µM $[Ca^{2+}]_o$ (low) and 2 mM $[Ca^{2+}]_o$ (normal) under otherwise identical conditions (*Figure 7A*). In agreement with indirect read-outs, bicarbonate did not change the cAMP concentration at 10 µM $[Ca^{2+}]_o$, whereas readjusting $[Ca^{2+}]_o$ to 2 mM rapidly elevated cAMP levels (*Figure 7A*). These results demonstrate that stimulation of SACY activity by bicarbonate requires $Ca^{2+}$ and that the regulation by $Ca^{2+}$ is fully and rapidly reversible.

Next, we studied basal SACY activity (in the absence of bicarbonate) when $Ca^{2+}$ was stepped to various lower $[Ca^{2+}]_o$ (*Figure 7C,D*). Surprisingly, stepping $[Ca^{2+}]_o$ from 2 mM to 20 µM moderately increased cAMP levels (*Figure 7C*). A similar increase was observed when $[Ca^{2+}]_o$ was decreased from 2 mM to 443 nM orfrom 10 µM to 2.2 nM (*Figure 7C*). Stimulation with bicarbonate at 2.2 nM $[Ca^{2+}]_o$ did not further enhance cAMP levels; however, when $[Ca^{2+}]_o$ was re-adjusted to 2 mM, bicarbonate was able to stimulate SACY activity (*Figure 7D*).

As a control, we analyzed whether changing $[Ca^{2+}]_o$ has an effect on the intracellular $pH_i$. Thus, we measured changes in pH using BCECF in wild-type mouse sperm after changing $[Ca^{2+}]_o$. Reducing $[Ca^{2+}]_o$ to 443 nM by addition of 3 mM BAPTA did not change $pH_i$ (*Figure 7F*). We conclude that the $\triangle[Ca^{2+}]_o$-evoked changes in FRET reflect changes in cAMP rather than in $pH_i$.

In summary, these results demonstrate that stimulation of SACY by bicarbonate requires $Ca^{2+}$ and that $Ca^{2+}$ alone, in the absence of bicarbonate, inhibits rather than activates SACY.

## Targeting mlCNBD-FRET to primary cilia

Although components of cAMP signaling are also localized to primary cilia (*Berbari et al., 2007*; *Bishop et al., 2007*; *Händel et al., 1999*), the physiological function of cAMP in primary cilia is ill defined. The analysis has been hampered by the lack of suitable tools to manipulate and analyze cAMP signaling in such a tiny cellular compartment. We set out to target mlCNBD-FRET to primary cilia. Protein import into the cilium is tightly regulated by the intraflagellar transport machinery (*Rosenbaum and Witman, 2002*). For targeting to cilia, we fused the mlCNBD-FRET sensor to the

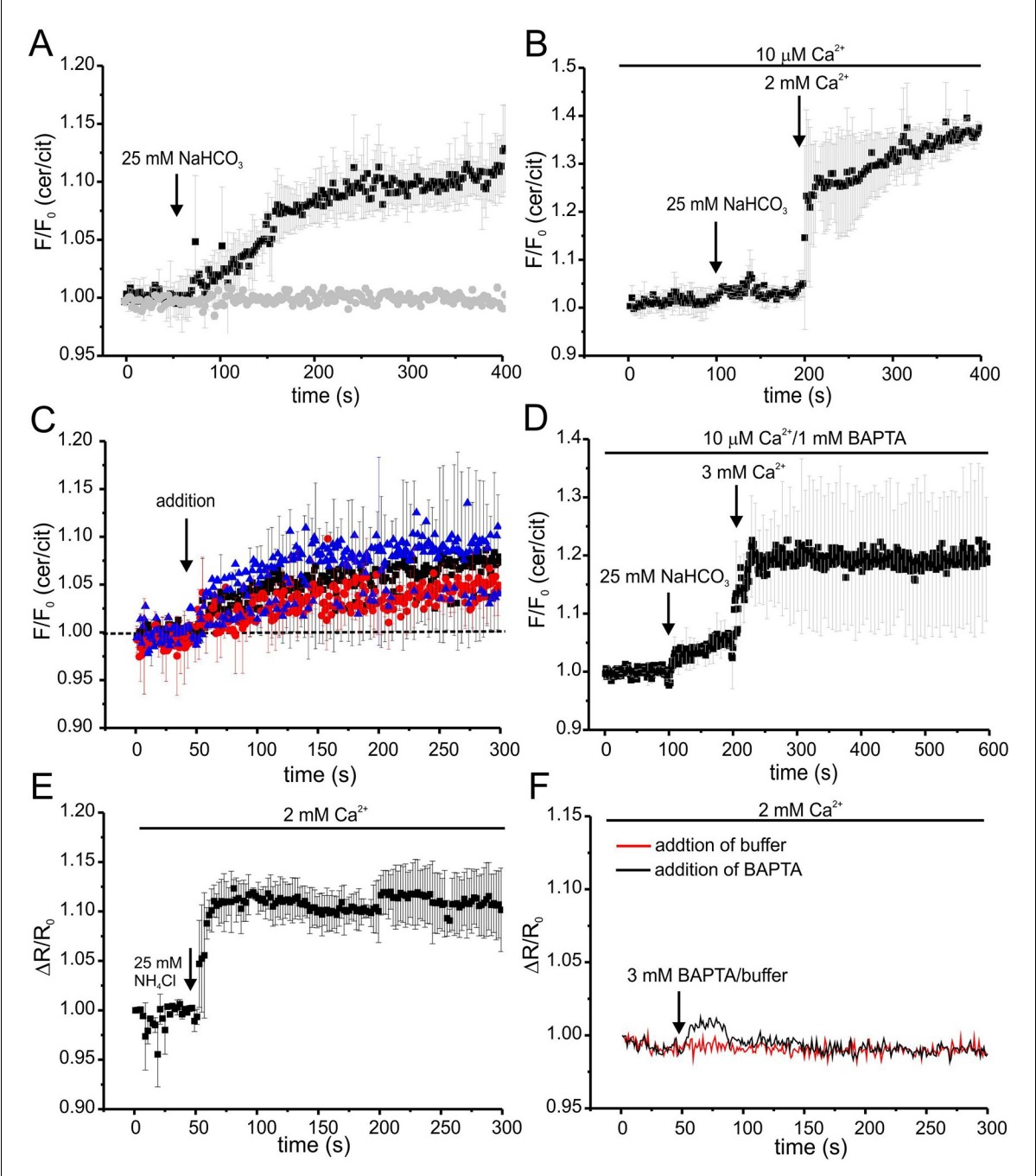

**Figure 7.** $Ca^{2+}$ regulation of cAMP dynamics in sperm. (**A**) Changes in FRET after stimulation of mlCNBD-FRET sperm with 25 mM bicarbonate. Sperm have been either kept in 2 mM (black) or 10 μM (grey) $Ca^{2+}$ buffer. FRET has been measured using a spectrofluorometer; n = 3 for each condition. (**B**) Changes in FRET after stimulation of a mlCNBD-FRET sperm kept 10 μM $Ca^{2+}$ buffer with 25 mM bicarbonate, followed by addition of 2 mM $Ca^{2+}$ (final concentration); n = 3. (**C**) Changes in FRET of mlCNBD-FRET sperm kept in 2 mM $Ca^{2+}$ buffer after the addition of 2 mM BAPTA (final: 20 μM $Ca^{2+}$, black), kept in 10 μM $Ca^{2+}$ buffer after the addition of 1 mM BAPTA (final: 2.2 nM $Ca^{2+}$, red), or kept in 2 mM $Ca^{2+}$ buffer after the addition of 3 mM BAPTA (final: 443 nM $Ca^{2+}$, blue). Arrow indicates the addition of $Ca^{2+}$ or BAPTA, the dotted line indicates the baseline; n = 4 for each condition. (**D**) Changes in FRET of mlCNBD-FRET sperm kept in 10 μM $Ca^{2+}$/1 mM BAPTA after addition of bicarbonate followed by 3 mM $Ca^{2+}$ (final: 2 mM $Ca^{2+}$). Data is shown as mean ± S.D; n = 4. (**E**) Changes in $pH_i$ of wild-type sperm kept in 2 mM $Ca^{2+}$ buffer after addition of 25 mM $NH_4Cl$. Data is shown as mean ± S.D.; n = 3. (**F**) Changes in $pH_i$ of wild-type sperm kept in 2 mM $Ca^{2+}$ buffer after addition of 3 mM BAPTA (final: 443 nM $Ca^{2+}$, black) or buffer (red) as a control. Data represents the mean of n = 3.

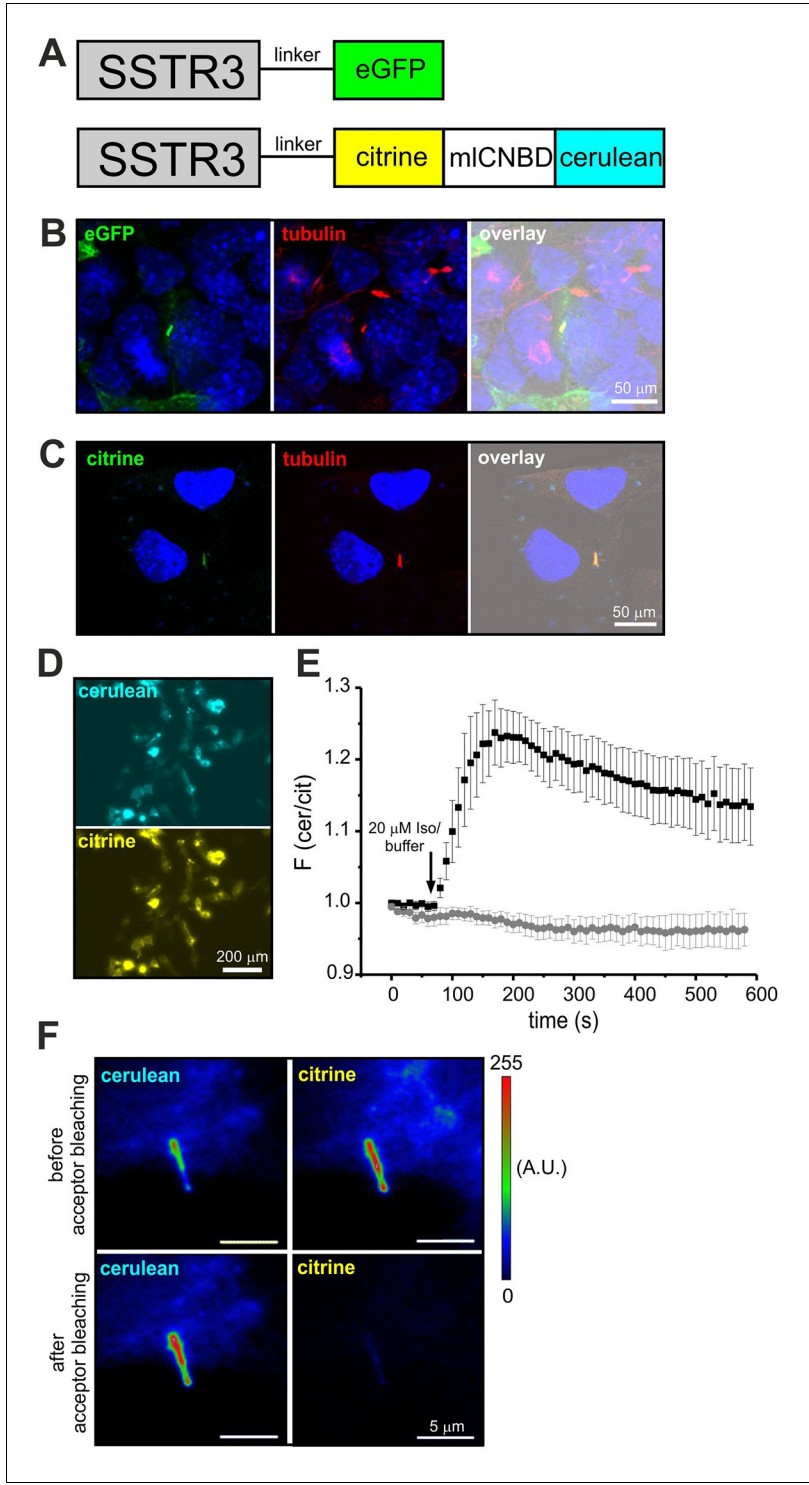

**Figure 8.** Targeting mlCNBD-FRET to primary cilia. (**A**) Strategy to target a protein to cilia. The somatostatin receptor 3 (SSTR3) has been fused to green fluorescent protein (eGFP) or mlCNBD-FRET. (**B**) Expression of eGFP in primary cilia of IMCD3 cells. An anti-acetylated tubulin antibody has been used as a marker for primary cilia. DNA has been labeled using DAPI. Scale bar is indicated. (**C**) Expression of mlCNBD-FRET in primary cilia of IMCD3 cells. Citrine fluorescence indicates the expression of mlCNBD-FRET. An anti-acetylated tubulin antibody has been used as a marker for primary cilia. DNA has been labeled using DAPI. Scale bar is indicated. (**D**) Representative image for HEK293 cells expressing SSTR3-mlCNBD-FRET. (**E**) Changes in FRET in HEK293 cells expressing SSTR3-mlCNBD-FRET (see **D**) after stimulation with 20 µM isoproterenol (black) or buffer only (grey). *Figure 8 continued on next page*

*Figure 8 continued*

FRET has been measured using fluorescence microscopy. Data are presented as mean ± S.D.; n = 9 for each condition. (**F**) Acceptor photobleaching. The citrine (acceptor) fluorescence of mlCNBD-FRET in IMCD3 cells was bleached for 2 min with a mercury lamp using a 510/20 nm filter. A representative image is shown. The cerulean emission was recorded before and after acceptor photobleaching. Relative fluorescence intensities are color-coded from low (blue) to high (red). Scale bars are indicated.

C-terminus of the ciliary somatostatin receptor 3 (SSTR3) (*Händel et al., 1999*); as control, we fused a green-fluorescent protein (eGFP) to SSTR3 (*Figure 8A*). Both constructs were expressed in IMCD3 cells, which carry primary cilia. In fact, both constructs were targeted to primary cilia (*Figure 8B,C*). To test whether fusion of mlCNBD-FRET to SSTR3 affects the sensor properties, SSTR3-mlCNBD-FRET was expressed in HEK293 cells. Compared to mlCNBD-FRET, SSTR3-mlCNBD-FRET was localized to intracellular membranes rather than uniformly distributed throughout cells (*Figure 3B vs. 8D*). Stimulation with isoproterenol (20 μM) increased the FRET ratio by approximately 25% compared to 40% for the non-tagged sensor (n = 9, *Figure 3E vs. 8E*). Thus, the SSTR3-mlCNBD-FRET fusion protein allows measuring cAMP dynamics. The functionality of mlCNBD-FRET in primary cilia of IMCD-3 cells was tested by acceptor photobleaching experiments. Upon bleaching, the cerulean emission increased (n = 11, 43 ± 21%), demonstrating that the SSTR3-fusion protein undergoes FRET (*Figure 8F*). In summary, mlCNBD-FRET is a versatile tool that can be used to measure cAMP dynamics with nanomolar sensitivity in solution, in different cell types, and small cellular compartments like the cilium or flagellum.

## Discussion

We developed a novel FRET-based biosensor to quantitatively measure cAMP dynamics. The sensor is quite versatile. First, unlike any other cAMP biosensor, the purified mlCNBD-FRET protein can be used in solution. Remarkably, the properties in solution and inside cells are by and large similar. Second, mlCNBD-FRET can be selectively targeted to subcellular compartments like primary cilia or sperm flagella to directly measure cAMP dynamics. Third, the sensor exhibits a significantly improved cAMP sensitivity compared to other single-protein biosensors, e.g. Epac-based FRET sensors. Thereby, it extends the range of accessible cAMP concentrations to the low nanomolar range. In fact, other cAMP sensors are unable to quantify resting cAMP levels in some cells (*Börner et al., 2011*). Finally, the sensor is compatible with fluorescence intensity and lifetime-based approaches. These favorable characteristics prompted us to revisit a lingering debate about cAMP signaling in sperm.

One issue concerns the presence, localization, and function of tmACs in sperm (*Brenker et al., 2012*; *Buffone et al., 2014*; *Vacquier et al., 2014*; *Wertheimer et al., 2013*); for a detailed account of pros and cons see (*Brenker et al., 2012*). SACY is the major source for cAMP in sperm and total cAMP levels in SACY$^{-/-}$ sperm are below the detection limit (*Xie et al., 2006*). Members of the tmAC family and the corresponding G proteins reportedly localize to the sperm head, i.e. they are spatially segregated from SACY and PKA in the flagellum (*Wertheimer et al., 2013*). Our result that bicarbonate, but not NKH477, increases flagellar cAMP levels strengthens the notion that SACY represents the only cAMP source in the flagellum. Furthermore, considering that the head volume is much larger than that of the flagellum, our results also argue that the flagellum either is sealed up and that cAMP in the head does not propagate to the flagellum, or that NKH477-sensitive tmACs are absent altogether. A physical barrier that restricts diffusion of cAMP between midpiece and head is unlikely to exist, because Ca$^{2+}$ entering the flagellum via CatSper channels propagates to the head (*Servin-Vences et al., 2012*; *Xia et al., 2007*). However, PDEs might serve as gatekeepers that prevent rapid cAMP exchange between head and flagellum.

Another conundrum concerns the relationship between cAMP signaling and tyrosine phosphorylation during capacitation. An important requisite for capacitation *in vitro* is bicarbonate and the ensuing rise of cAMP followed by PKA activation. PKA activity must be maintained for at least 30 min to initiate tyrosine phosphorylation (*Morgan et al., 2008*), which requires 90 min to complete. However, in mice, total cAMP levels return to basal values within 10 min (*Figure 5D*). This begs the

question how PKA activity is sustained when cAMP levels recover during bicarbonate stimulation. We resolve this conundrum in mouse sperm by showing that the free cAMP concentration stays up during measurements (*Figure 5D*). Thus, total cAMP levels do not properly reflect free cAMP concentrations in live cells. This might be due to compartmentalization of cAMP signaling in sperm, similar to what has been shown for cardiomyocytes (*Zaccolo et al., 2006*), or due to buffering by cAMP-binding proteins. A case in point is the rod photoreceptor (*Yau, 1994*). The free cGMP concentration in the dark is approximately 1 µM, whereas most of the total cGMP content (50 µM) is bound to high-affinity sites of PDEs (*Cote et al., 1984*; *1986*). Light stimulation activates PDE, cGMP levels drop, and cyclic nucleotide-gated ion channels close, giving rise to a hyperpolarizing light response. Notably, a drop of the total cGMP concentration (maximally 50% ) requires light intensities that are orders of magnitude larger than those that saturate the light response (*Cote et al., 1984*; *1986*). Likewise, the total cGMP concentration in sea urchin sperm is a few µM, whereas, at rest, the free cGMP must be $\leq$3 nM (*Bönigk et al., 2009*; *Kaupp et al., 2003*). The $Ca^{2+}$ response in sea urchin sperm saturates at $\leq$20 pM of the chemoattractant, whereas approximately 5 nM chemoattractant are required to detect a noticeable rise of total cGMP concentration (*Kaupp et al., 2003*).

A third issue concerns the regulation of SACY by $Ca^{2+}$. SACY activity is stimulated by bicarbonate in buffer containing 2 mM $Ca^{2+}$, but not in low micromolar $Ca^{2+}$ buffer (*Figure 7A*) (*Carlson et al., 2007*; *Navarrete et al., 2015*; *Wennemuth et al., 2003*). Recent findings suggest that $Ca^{2+}$ has a biphasic role in the regulation of cAMP signaling (*Navarrete et al., 2015*): at micromolar $Ca^{2+}$-conditions, bicarbonate does not stimulate PKA activity or increase tyrosine phosphorylation; however, chelating $[Ca^{2+}]_o$ using EGTA in the presence of bicarbonate relieves the inhibition, stimulating PKA activity, and increasing tyrosine phosphorylation. Similarly, in CatSper1$^{-/-}$ sperm, $[Ca^{2+}]_i$ is lower than in wild-type sperm (*Ren et al., 2001*), yet tyrosine phosphorylation in the presence of bicarbonate is enhanced (*Chung et al., 2014*). One interpretation of these seemingly puzzling results suggests that bicarbonate can stimulate SACY at normal and very low $[Ca^{2+}]_i$, but not at intermediate $Ca^{2+}$ levels (*Navarrete et al., 2015*). However, such a biphasic control by $Ca^{2+}$ has not been observed *in vitro*. Instead, the activity of isolated SACY protein upon bicarbonate stimulation is steadily enhanced by $Ca^{2+}$ in a dose-dependent manner (*Jaiswal and Conti, 2003*). Alternatively, $Ca^{2+}$ might control cellular signaling downstream of cAMP. Here, we show that cAMP levels do not significantly change during bicarbonate stimulation at very low $Ca^{2+}$ concentrations (2 nM); thus, a bicarbonate-dependent cAMP increase is not underlying the stimulation of PKA activity and tyrosine phosphorylation under these conditions. However, under conditions that are expected to significantly lower $[Ca^{2+}]_i$, basal SACY activity is stimulated, indicating that SACY activity is controlled by a $Ca^{2+}$-dependent negative feedback that operates in intact sperm, but is absent on the isolated or heterologously expressed protein. Regulation of SACY activity via another negative feedback has been described before (*Burton and McKnight, 2007*; *Nolan et al., 2004*). In sperm lacking the catalytical PKA subunit C$\alpha$2, both basal and bicarbonate-stimulated cAMP levels are higher than in wild-type sperm (*Burton and McKnight, 2007*; *Nolan et al., 2004*), suggesting that PKA-dependent phosphorylation indirectly or directly down-regulates SACY activity.

cAMP-signaling components are not uniformly distributed throughout sperm: PKA and SACY are exclusively localized to the flagellum (*Hess et al., 2005*; *Nolan et al., 2004*; *Wertheimer et al., 2013*). However, even along the flagellum, cAMP signaling is compartmentalized; the cAMP dynamics stimulated by bicarbonate is different in the midpiece and the principal piece. We favor the idea that distribution and activity of PDEs is different in the midpiece and principal piece. PDE are targeted to distinct domains in sperm (*Bajpai et al., 2006*; *Lefièvre et al., 2002*). Moreover, at least two PDE isoforms and splice variants, PDE4 (cAMP-specific) and PDE1 ($Ca^{2+}$/calmodulin-dependent) are involved in the control of sperm motility and tyrosine phosphorylation (*Fisch et al., 1998*; *Leclerc et al., 1996*; *Lefièvre et al., 2002*; *Visconti et al., 1995*). Using mlCNBD-FRET mice, it will now be possible to analyze cAMP dynamics in different subcellular compartments and unravel the mechanisms underlying the differential regulation of cAMP signaling.

Finally, primary cilia constitute a unique $Ca^{2+}$ compartment that is functionally distinct from the cytoplasm of the cell body (*Delling et al., 2013*; *DeCaen et al., 2013*). $Ca^{2+}$ channels (PKD-Like1/2) in the plasma membrane of primary cilia maintain a higher resting $Ca^{2+}$ concentration compared to the cell body, despite steady $Ca^{2+}$ efflux via the ciliary base (*Delling et al., 2013*; *DeCaen et al., 2013*). Considering that primary cilia also host cAMP-signaling components (*Johnson and Leroux, 2010*), we hypothesize that cAMP signaling in primary cilia might also be functionally distinct from

the cell body. The combination of sensitive genetically-encoded cAMP biosensors and light-driven actuators (*Jansen et al., 2015*) provides the high spatial and temporal resolution that is needed to unravel cAMP-signaling pathways in cilia and flagella.

## Material and methods

### Cloning of mlCNBD-FRET

The mlCNBD cDNA sequence (cyclic nucleotide binding-domain) of the cyclic nucleotide-gated $K^+$ channel from *Mesorhizobium loti* (MAFF303099, mll3241) was amplified via PCR (*Cukkemane et al., 2007*). For expression in mammalian cell lines, citrine and cerulean were amplified by PCR and fused to the N- and C-terminus of the mlCNBD via *BamHI/HindIII* or *XhoI/ApaI*, respectively. The C-terminus of cerulean contained a histidine ($His_{10}$) tag. The PCR product was cloned into a pcDNA3.1(+) vector (Invitrogen, Darmstadt, Germany) using *BamHI* and *ApaI* (pc3.1-mlCNBD-FRET). For expression in *E.coli*, citrine was amplified by PCR from pc3.1-mlCNBD-FRET and an *NdeI* site was added at the 5'end to allow in-frame cloning into the pET21a vector (Novagen, Darmstadt, Germany). The mlCNBD fused to cerulean containing a $His_{10}$ tag was amplified by PCR from pc3.1-mlCNBD-FRET and an *EcoRI* site was added at the 3'end. PCR products were cloned into pET21a using *NdeI/EcoRI* (pET21a-mlCNBD-FRET). The mutation encoding R307Q was introduced using the QuikChange site-directed mutagenesis protocol (Agilent Technologies, Santa Clara, CA).

### Protein purification

The mlCNBD-FRET protein was expressed in *E. coli* (BL21 DE3 pLysE). Bacteria were grown at 37°C until density reached an $OD_{600}$ = 0.6. Then, expression of mlCNBD-FRET was induced using 0.6 mM IPTG. The cells were grown overnight at 20°C and harvested after 18 hr. The cells were lysed in lysis buffer (20 mM NaP, 500 mM NaCl, 10 mM DNAse, pH 7.4) using ultrasonification followed by centrifugation at 45,000 g, 20 min, and 4°C. The supernatant containing the mlCNBD-FRET protein was loaded onto a 1 ml HiTrap cobalt-IMAC column (GE) pre-equilibrated with the binding buffer (20 mM NaP, 500 mM NaCl, 10 mM imidazole, pH 7.4) in the Äkta HPLC system (GE Healthcare, Solingen, Germany). After loading, the column was washed with 5 column volumes of binding buffer. Bound cAMP was removed from the protein by incubation for 10 min with 1 ml 8-CPT-cGMP (5 mM) (Biolog, Bremen, Germany). This step was performed three times followed by a washing step with 10 ml binding buffer. The mlCNBD-FRET was then eluted using a linear gradient (0–100%) of elution buffer (20 mM $Na_3PO_4$, 500 mM NaCl, 500 mM imidazole, pH 7.4).

To enrich properly folded protein, gel-filtration chromatography of mlCNBD-FRET was carried out using a Superdex 200, Hi-Load 16/60 column (Amersham Biosciences/GE Healthcare), which allows separating proteins with molecular weight between 30 and 200 kDa. The mlCNBD-FRET protein has an apparent molecular weight of 70.9 kDa. The column was washed and equilibrated with 2 column volumes of the gel-filtration buffer (GF buffer, 10 mM $K_2HPO_4$, 100 mM KCl, pH 7.4). Subsequently, the sample was injected onto the column and protein elution was followed at 280 nm. The protein fractions corresponding to the monomer were collected. This sample was flash frozen in GF buffer containing 10% glycerol and stored at -80°C.

### Determination of cyclic nucleotide content by HPLC

The purified mlCNBD-FRET still contained cAMP (*Cukkemane et al., 2007*). The cAMP was removed by incubation with the low-affinity analog 8-CPT-cGMP (*Cukkemane et al., 2007*) followed by extensive washing. To determine the cAMP content, protein samples were denatured and analysed as described before (*Cukkemane et al., 2007*) using a Zorbax SB-C18 column (5 μm, 250 mm x 0.46 mm, Agilent, Waldbronn, Germany). The column was calibrated and the detection limit for cAMP was determined by injecting increasing amounts of cAMP prepared in GF buffer. A linear gradient was produced from solution A (5 mM $KH_2PO_4$, pH 5.0) and solution B (80% methanol) at a flow rate of 0.3 ml/min. The absorbance for cAMP was detected at 256 nm with a retention time of 6.3 min. The final cAMP content of mlCNBD-FRET was below the detection limit.

## Ligand-binding assay

To determine the affinity of the purified mlCNBD-FRET, known protein concentrations ($\varepsilon$ = 53,290 $M^{-1}cm^{-1}$, MW = 70.9 kDa, calculated by ProtParam tool) were mixed with increasing concentrations of cAMP (Sigma-Aldrich, Seelze, Germany). The fluorescence was observed in a spectrofluorometer (Quantamaster 40, PTI) with excitation at 430 nm (cerulean) and emission at 471 nm (cerulean) and 529 nm (citrine). Fluorescence spectra were recorded using 430 nm excitation. FRET was calculated from the ratio of the emission intensities for cerulean at 471 nm over citrine at 529 nm. The change in FRET $\triangle$F was plotted versus the cAMP concentration and fitted with a substrate-depletion model (*Cukkemane et al., 2007*) to calculate the $K_D$ value for cAMP.

To determine the effect of pH on the mlCNBD-FRET sensor, the protein was exposed to different pH solutions (pH 6–8). The buffer solutions consisted of 100 mM KCl/10 mM HEPES and the pH was adjusted using 1 M NaOH. The protein was kept in 100 mM KCl/10 mM HEPES pH 7.4. For measurements, the protein solution (50 µM stock) was mixed 1:1000 with the corresponding pH buffer and the final pH of the solution was controlled again using a pH meter. Fluorescence spectra were recorded in a spectrofluorometer (Quantamaster 40, PTI) with excitation at 430 nm. The binding affinity was determined as described for the ligand-binding assay.

## Kinetic measurements using the stopped-flow technique

For time-resolved recordings of FRET changes, purified mlCNBD-FRET protein was mixed 1:1 with cAMP in GF buffer in a rapid-mixing device (SFM400, Bio-Logic) at a flow-rate of 9.5 ml/s. Cerulean was excited by the blue LED of a SpectraX Light Engine (Lumencor, Beaverton, OR) and the excitation light was passed through a 435/24 nm filter (Semrock Brightline, Rochester, NY) onto the µFC08 cuvette. The emission was collected through 485/15 nm (cerulean) and 535/15 nm filters (citrine, Semrock Brightline). Fluorescence emission was recorded by photomultiplier modules (9656–20; Hamamatsu Photonics, Herrsching, Germany). Data acquisition was performed at 200 Hz with a data acquisition pad (PCI-6221; National Instruments, Austin, TX) and the Bio-Kine software v. 4.49 (Bio-Logic). Numerical analysis was carried out with Dynafit (v 3.28.070, BioKin, Watertown, MA) using a simple binding scheme.

$$E \,+\, L \underset{K_{off}}{\overset{K_{on}}{\rightleftarrows}} EL$$

E, L, EL refer to the concentrations of the receptor, ligand, and receptor-ligand complex, respectively. $k_{on}$ describes the *on* rate for ligand binding along with the associated conformational change resulting in a FRET signal, $k_{off}$ describes the *off* rate of ligand dissociation from the receptor and the associated return to its resting-state conformation.

## Measuring cAMP dynamics in mammalian cell lines

HEK293 cells (ATCC CRL-1573, authentication method: STR profiling, mycoplasma test negative) were electroporated with pc3.1-mlCNBD-FRET or pc3.1-mlCNBD-FRET-R307Q using the Neon 100 µl kit (Invitrogen, Darmstadt, Germany) and the MicroPorator (Digital Bio, Invitrogen) according to the manufacture's protocol (3x 1245 mV pulses with 10 ms pulse width). Cells were transferred into complete medium composed of DMEM plus GlutaMax (Life Technologies GmbH, Carlsbad, CA) and 10% fetal bovine serum (Biochrom, Berlin, Germany). For the selection of monoclonal HEK293 cells stably expressing mlCNBD-FRET, the antibiotic G418 (800 µg/ml, Invitrogen) was added 24 hr after electroporation. Monoclonal cell lines were identified by Western blot and immunocytochemistry.

Cells were cultured in Ibidi µ-slides (Ibidi, Planegg, Germany) attached to a home-built gravity-flow perfusion system. Measurements were performed using a CellR live-cell imaging system (Olympus) with 20x objective. Cells were perfused with extracellular solution (ES: 120 mM NaCl, 5 mM KCl, 2 mM $MgCl_2$, 2 mM $CaCl_2$, 10 mM HEPES, 10 mM glucose, pH 7.4) to determine the basal FRET ratio and then the stimulus was added (NKH477 and IBMX, Tocris, Wiesbaden, Germany; Isoproterenol and SNP, Sigma-Aldrich). FRET was recorded by exciting cerulean at 436 nm and measuring the emission of cerulean and citrine at 470 nm and 535 nm, respectively. The change in FRET (emission$_{cerulean}$/emission$_{citrine}$) was calculated, corrected for bleed-through, normalized to the baseline, and plotted versus time.

For population measurement using a spectrofluorometer (Quantamaster 40, PTI), HEK293 cells were trypsinized at 37°C for 5 min and washed twice with ES. Measurements were performed in PMMA cuvettes at $1\times10^5$ cells/ml under constant stirring (Spinbar, Bel-art products, Wayne, NJ). Fluorescence spectra at steady-state were recorded with excitation at 430 nm and 0.5 s integration time. For time-resolved measurements, cells were excited at 430 nm and emission was recorded at 470 nm (cerulean) and 530 nm (citrine) at an acquisition frequency of 1 Hz. The change in FRET ($emission_{cerulean}/emission_{citrine}$) was calculated, corrected for bleed-through, normalized to the baseline, and plotted versus time.

For repetitive stimulation experiments, cells were stimulated with 500 nM isoproterenol and fluorescence was measured in a plate reader (FLUOstar Omega; BMGLabtech) at 29°C with excitation at 440 nm and emission at 485 and 520 nm. The wash-out was performed by removing the stimulus and addition of ES. The fluorescence emission ratio was normalized to the initial baseline fluorescence ratio and plotted as a function of time.

## Generation and genotyping of *Prm1*-mlCNBD-FRET transgenic mice

A hemagglutinin (HA) tag was fused to the C-terminus, an *EcoR*I restriction site was added to the 5' end, the internal *BamH*I site was deleted, and a *BamH*I and *Xba*I restriction site was added to the 3' end by PCR. The PCR product was cloned into a pBluescript SK- vector (Agilent Technologies, Santa Clara, USA) using *EcoR*I and *Xba*I (pB-mlCNBD-FRET). After sequencing, the mlCNBD-FRET-HA insert was excised and cloned into pPrCExV (kind gift from Robert Braun, Jackson Laboratory) using *BamH*I (pPrCExV-mlCNBD-FRET) to express mlCNBD-FRET under the control of the protamine-1 promotor that is exclusively active in post-meiotic spermatids (*Zambrowicz et al., 1993*). Transgenic mice were generated via pronuclear injection following standard procedures (*Ittner and Götz, 2007*) at the transgenic facility of the LIMES (University of Bonn, Germany; licence number: 84–02.04.2012.A192). Mice were genotyped by PCR using mlCNBD-FRET-specific primers (#1: 5'-GTACAAGGGTACCCAAGAAGTCCGTCGC-3', #2: 5'-CGAAGCACTGCACGCCCCAGGTC-3'). Mice used in this study were 2–5 months of age. All animal experiments were in accordance with the relevant guidelines and regulations.

## Western blot analysis

Protein lysates were obtained by homogenizing cells or tissue in lysis buffer (10 mM Tris/HCl, pH 7.6, 140 mM NaCl, 1 mM EDTA, 1% Triton X-100, mPIC protease inhibitor cocktail 1:500) followed by trituration through a 18-gauge needle. Samples were incubated for 30 min on ice and centrifuged at 10,000 g for 5 min at 4°C. Prior to separation by SDS-PAGE, samples were mixed with 4x SDS loading-buffer (200 mM Tric/HCl, pH 6.8, 8% SDS (w/v), 4% β-mercaptoethanol (v/v), 50% glycerol, 0.04% bromophenol blue) and heated for 5 min at 95°C. Sperm samples used for SDS-PAGE were washed with 1 ml PBS and sedimented by centrifugation at 5000 g for 5 min. 1–2 x $10^6$ cells were resuspended in 4 x SDS loading buffer and heated for 5 min at 95°C. For Western-blot analysis, proteins were transferred onto PVDF membranes (Merck Millipore, Billerica, USA), probed with antibodies, and analysed using the Odyssey Imaging System (LI-COR, Lincoln, NE) detection-system.

Primary antibodies: anti-HA 3F10 (1:5000; Roche, Basel, Switzerland), anti-α-tubulin (1:5000; Sigma-Aldrich) anti-GFP antibody (1:5000; abcam, Cambridge, UK); secondary antibodies: IRDye680 and IRDye800 antibodies (1:20,000, LI-COR); ICC: fluorescently-labeled antibodies (1:500; Dianova, Hamburg, Germany).

## Immunohistochemistry

Testes were fixed in 4% paraformaldehyde/PBS overnight, cryo-protected in 10 and 30% sucrose, and embedded in Tissue-Tek (Sakura Finetek, Alphen aan den Rijn, Netherlands). To block unspecific binding sites, cryosections (16 µm) were incubated for 1 hr with blocking buffer (0.5% Triton X-100 and 5% ChemiBLOCKER (Merck Millipore) in 0.1 M NaP, pH 7.4). The primary anti-HA 3F10 antibody (rat monoclonal; Roche, Basel, Switzerland) was diluted 1:1000 in blocking buffer and incubated for 2 hr. Fluorescent secondary antibodies (donkey anti-rat CY3; Dianova) were diluted 1:500 in blocking buffer containing 0.5 mg/ml DAPI (Life Technologies) and pictures were taken on a confocal microscope (FV1000; Olympus).

## Sperm preparation

Sperm were isolated by incision of the cauda followed by a swim-out in modified TYH medium (135 mM NaCl, 4.8 mM KCl, 2 mM $CaCl_2$, 1.2 mM $KH_2PO_4$, 1 mM $MgSO_4$, 5.6 mM glucose, 0.5 mM sodium pyruvate, 10 mM L-lactate, 10 mM HEPES, pH 7.4). For capacitation, the medium contained 3 mg/ml BSA and 25 mM of NaCl was substituted with 25 mM $NaHCO_3$. The pH was adjusted at 37°C. After 15–30 min swim-out at 37°C, sperm were collected and counted.

## Fluorescence spectroscopy to measure cAMP dynamics in sperm populations

After swim-out, sperm were kept in 500 µl tubes filled up with TYH buffer to avoid pH changes due to dissolution of ambient $CO_2$ into the buffer. Measurements were performed in TYH buffer containing 1–5 x $10^4$ sperm/ml. The fluorescence recording was performed at a spectrofluorometer (Quantamaster 40, PTI) at 37°C as described for the mammalian cells lines.

Nominally $Ca^{2+}$-free TYH buffer was prepared by adding 10 µM $Ca^{2+}$ to TYH buffer prepared without any $Ca^{2+}$. When adding 25 mM $NaHCO_3$ (1:10 from 250 mM stock), the $Na^+$ concentration in modified TYH was reduced to 110 mM NaCl. BAPTA (1,2-Bis(2-aminophenoxy)ethane-N,N,N',N'-tetraacetic acid tetrapotassium salt, Sigma-Aldrich) was dissolved in the modified TYH buffer (pH 7.4) and added to sperm samples at a final concentration of 1, 2, or 3 mM.

## Intracellular pH measurement with BCECF

Cells were loaded with 10 µM (HEK293 cells) or 5 µM BCECF-AM (sperm) (Invitrogen) for 10 min at 37°C and 10% $CO_2$. Afterwards, HEK293 cells were washed three times in 1 ml ES before starting the measurement. For sperm, the dye was removed by single centrifugation (700 x g, 7 min, RT) and resuspension in TYH buffer. For HEK293 cells, fluorescence was measured in a plate reader (FLUOstar Omega; BMGLabtech) at 29°C. For calibrating the intracellular pH ($pH_i$) using the null-point method, cells were permeabilized with Triton-X 100 (final concentration of 0.1% ) followed by addition of different pH buffer solutions (pH 5–8). BCECF was excited at 440 nm and 485 nm and the emission was detected at 520 nm. The fluorescence emission ratio at both excitation channels was normalized to the initial fluorescence ratio and plotted as a function of pH. A linear regression analysis (Origin; OriginLab) was performed to calculate the $pH_i$.

Changes in $pH_i$ in sperm were either analysed in a rapid-mixing device (SFM-400; Biologic) in the stopped-flow mode or in a spectrofluorometer (Quantamaster 40, PTI) at 37°C. In the stopped-flow device, the sperm suspension ($5 \cdot 10^6$ sperm/ml) was rapidly mixed 1:1 (v/v) at a flow rate of 0.5 ml/s with the respective stimulants. Fluorescence was excited by a SpectraX Light Engine (Lumencor), whose intensity was modulated with a frequency of 10 kHz. The excitation light was passed through a BrightLine 452/45 nm filter (Semrock) onto the cuvette (FC-15, Biologic). Emission light was recorded in a dual-emission mode using BrightLine 494/20-nm and BrightLine 540/10-nm filters (Semrock) by photomultiplier modules (H10723-20; Hamamatsu Photonics). The signal was amplified and filtered through a lock-in amplifier (7230 DualPhase, Ameteky). Data acquisition was performed with a data acquisition pad (PCI-6221; National Instruments) and Bio-Kine software v. 4.49 (Bio-Logic, Illingen, Germany). The pH signal represent the ratio of F494/540 and is depicted as the percent of the relative change in ratio ($\Delta R/R$) with respect to the mean of the first three data points at the onset of the signal. The control (TYH) signal was subtracted from the $NH_4Cl$ or bicarbonate traces.

To analyze the effect of changing the extracellular $Ca^{2+}$ on the intracellular pH, measurements were performed at the spectrofluorometer in respective TYH buffer containing 1 x $10^4$ sperm/ml. BAPTA (1,2-Bis(2-aminophenoxy)ethane-N,N,N',N'-tetraacetic acid tetrapotassium salt, Sigma-Aldrich) was dissolved in the modified TYH buffer (pH 7.4) and added to the sperm samples at a final concentrations of 3 mM. BCECF was excited at 452 nm and the emission was detected at 494 nm and 540 nm. The fluorescence emission ratio of 540/494 was calculated and plotted against time. The ratio was corrected for a drift in fluorescence. The drift was determined by calculating the slope of the response after addition of buffer experiment, which was multiplied to the time axis to calculate the drift over time. The drift was subtracted from the sample data. The drift corrected data was finally normalized to the initial emission ratio.

## Determination of total cAMP content

After stimulation with 25 mM NaHCO$_3$, the reaction was quenched with HClO$_4$ (1:3 (v/v); 0.5 M final concentration). Samples were frozen in liquid N$_2$, thawed, and neutralized by addition of K$_3$PO$_4$ (0.24 M final concentration). The salt precipitate and cell debris were sedimented by centrifugation (15 min, 15,000 g, 4°C). The cAMP content in the supernatant was determined by a competitive immunoassay (Molecular Devices, Sunnyvale, CA) including an acetylation step for higher sensitivity. Calibration curves were obtained by serial dilutions of cAMP standards. 96-well plates were analysed by using a microplate reader (FLUOstar Omega; BMGLabtech).

## Calibration of the FRET sensor in HEK293 cells

To determine the basal concentration of cAMP in HEK293 cells, the null-point method was used. HEK293 cells stably expressing mlCNBD-FRET were trypsinized and cells were suspended in 1 ml ES buffer (5 x 10$^5$ cells/ml). The fluorescence was recorded for 5 min in a spectrofluorometer (Quanta-master 40, PTI) with excitation at 430 nm and emission at 471 nm (cerulean) and 529 nm (citrine). Cells were then permeabilized with 20 µM digitonin to deplete intracellular cAMP and the concomitant FRET signal was measured. During permeabilization, the pH$_i$ might change if the extracellular and intracellular pH is different. Using the pH indicator BCECF, we determined a pH$_i$ of 7.2 ± 0.1 (n = 4) in mlCNBD-FRET-expressing HEK293 cells. Accordingly, we set pH$_o$ during permeabilization to 7.2 to assure that no pH change occurs. Increasing concentrations of cAMP were then added to the cell suspension that eventually saturate the FRET sensor. The FRET ratio for each cAMP concentration was plotted against the cAMP concentrations and analysed by linear regression. The basal cAMP concentration was defined as the external cAMP concentration where no change in FRET was observed.

## Multi-photon fluorescence lifetime microscopy (FLIM)

The decay of the fluorescence lifetime was measured on a Trimscope II (LaVision BioTec, Göttingen, Germany) equipped with a FLIM x16 TCSPC detector. Cells were imaged using a 20x NA = 1.0 water-dipping objective and excited with an 80 MHz mode-locked laser tuned to 810 nm. Emitted photons were filtered through a 700 nm short-pass filter followed by a 495 nm dichroic mirror that reflected the fluorescence below 488 nm towards the FLIM detector. Images were acquired with 512x512 pixels, pixel size = 800 nm. The instrument response function (IRF) was obtained from second harmonic signals generated from a urea sample at the same excitation wavelength. Cells were imaged in ES in culture dishes at 23°C. To increase the intracellular cAMP concentration, cells were stimulated for 5 min using 40 µM NKH/500 µM IBMX. As a control, 0.05% DMSO was used. To permeabilize the cells and reduce the intracellular cAMP concentration, cells were treated with 20 µM digitonin for 10 min. To measure the maximal response, cells were incubated with 5 µM cAMP. Time constants of fluorescence decay were calculated using the FLIMfit software tool (*Warren et al., 2013*). The fitting function consisted of a double exponential decay model that was convolved with the IRF. Pixels below 20 counts were excluded from the analysis. 15x15 pixels were then binned to increase photon count and improve the accuracy of the fit. The analysis generated two time constants ($\tau_1$ and $\tau_2$) associated with the double exponential decay. Individual amplitudes ($a_1$ and $a_2$) reflecting the magnitude of each decay constant were used to calculate the weighted average $\tau_{average}$ for every pixel using the following equation:

$$\tau_{average} = \frac{a_1 \times \tau_1 + a_2 \times \tau_2}{a_1 + a_2}$$

## Fluorescence lifetime spectroscopy

Decay curves were recorded by Time-Correlated Single-Photon Counting (TCSPC) using the Fluo-Time 300 (PicoQuant). Cells in solution (10$^6$ cells/ml in ES) or purified protein (1 µM) were measured in a cuvette and excited by a pulsed diode laser at 440 nm (PicoQuant) and fluorescence emission was detected at 480/20 nm. The IRF was obtained using Ludox suspension. The cerulean lifetime decay was measured for cells in ES before stimulation with 40 µM NKH/500 µM IBMX. Cells were suspended in 1 ml ES in a cuvette. To calibrate mlCNBD-FRET in cells using FLS, the decay of the cerulean lifetime was first recorded in ES followed by the addition of 20 µM digitonin to

permeabilize the cells and then by addition of increasing cAMP concentrations. Fitting was performed using the PicoQuant FluoFit software. A double-exponential function was chosen for the analysis with iterative reconvolution using the IRF obtained from Ludox.

### Cloning ciliary constructs

The full-length mouse SSTR3 cloned into pEGFP-N1 was kindly provided by Greg Pazour (UMass Medical School). For expression of SSTR3-tagged mlCNBD-FRET, the SSTR3 coding sequence was amplified by a recombinant PCR reaction to delete the internal *NheI* site. The fragment was subcloned into pc3.1-mlCNBD-FRET vie *NheI/HindIII*.

### Acceptor photobleaching

During acceptor photobleaching, the fluorescence of the FRET donor is recorded while bleaching the FRET acceptor. IMCD-3 cells (ATCC CRL-2123, authentication method: STR profiling, mycoplasma test negative) were transfected with pc3.1-SSTR3-mlCNBD-FRET using the Neon 100 µl kit (Invitrogen) and the MicroPorator (Digital Bio) according to the manufacture's protocol. Afterwards, cells were plated on coverslips and the experiment was performed on an inverted microscope (eclipse Ti, Nikon, Düsseldorf, Germany), equipped with EMCCD camera (iXON Ultra 897, Andor, Berlin, Germany), using 100x oil objective (CFI Apo TIRF, NA 1.49, Nikon). First, an image of the cilium was obtained in the donor channel using 405 nm laser excitation and 480/40 nm emission and an image in the acceptor channel using 488 nm laser excitation and 560/40 nm emission. Photobleaching was performed at 510/20 nm using a mercury lamp (C-LHGFIE, Nikon) for 2 min. This was followed by the acquisition of two images, one in the donor channel and another in the acceptor channel. The donor channel was compared to the images before and after acceptor photobleaching and the increase in intensity was quantified after background subtraction. The amount of bleaching was assessed by comparing citrine fluorescence before and after bleaching.

### FRET measurement using dual-view imaging

For spatial-temporal-resolved measurements, sperm from transgenic mice were imaged with a dual-view microscopic setup. Cerulean was excited at 440/20 nm (Lumencor Spectra X light engine) combined with a 438/24 nm excitation filter (Semrock). In addition, a dichroic mirror 455 nm DRLP and an additional filter with 460 nm ALP (Edmund optics, Karlsruhe, Germany) was used. Emission light was collected with an objective (40x, NA 0.95; Olympus, USA) and detected with an EMCCD Camera (iXon Ultra 897, Andor). In the dual-view emission path, a beam splitter (DV2, 515 dxcr, Mag Biosystems, Exton, PA) with a cerulean and citrine emission filter (475/28 nm and 535/50 nm, respectively; Semrock) was used. Thereby, both channels were acquired simultaneously with an acquisition frequency of 1 Hz. The resulting video was analyzed with a self-developed image analysis plugin for ImageJ. First, the citrine and the cerulean channel was registered (MultiStackReg, ImageJ). Second, the flagellum was tracked and the signal intensity of the citrine and the cerulean channel was determined along the flagellum. By using the pre-determined bleed-through value for the citrine channel ($0.69 \pm 0.01$), changes in FRET (($\text{emission}_{cerulean} - 0.69 \times \text{emission}_{citrine}$)/$\text{emission}_{citrine}$) was calculated, normalized to the baseline, and plotted versus time for each position along the arc length. The droplet separates the midpiece and the principal piece; thus, both regions were selected at a distance 5–10 µm apart from the droplet. The mean signal intensity and the standard deviation was determined in both regions for 20 µm along the arc length. The values were plotted over time and a logistic regression (Origin 9.0) was used to determine the kinetics within the two regions. The parameters of the logistic regression for different experiments were used to determine the average kinetics within the two regions.

## Acknowledgements

We thank JH Krause, R Pascal, J Hierer, D Herborn, and I Lux for technical assistance, W Bönigk for cloning, H-G Körschen for his guidance performing measurements at the FLUOstar, Greg Pazour (University of Massachusetts) for providing SSTR3-eGFP, R Braun (Jackson Laboratory) for providing the pPrCExV vector, and the transgenic service at LIMES for generating the transgenic mice.

# Additional information

## Funding

| Funder | Grant reference number | Author |
|--------|------------------------|--------|
| Deutsche Forschungsgemeinschaft | Bonn Excellence Cluster ImmunoSensation | Luis Alvarez U Benjamin Kaupp Dagmar Wachten |
| Fritz Thyssen Stiftung | Project Grant | Dagmar Wachten |
| Deutsche Forschungsgemeinschaft | SFB645 (INST 217/555-2) | Timo Strünker U Benjamin Kaupp Dagmar Wachten |
| Deutsche Forschungsgemeinschaft | Project Grant | Vera Jansen Dagmar Wachten |

The funders had no role in study design, data collection and interpretation, or the decision to submit the work for publication.

## Author contributions

SM, VJ, JFJ, Acquisition of data, Analysis and interpretation of data, Drafting or revising the article; HH, MD, MB, Acquisition of data, Analysis and interpretation of data; LA, Conception and design, Analysis and interpretation of data, Drafting or revising the article; TS, RS, UBK, Conception and design, Drafting or revising the article; DW, Conception and design, Acquisition of data, Analysis and interpretation of data, Drafting or revising the article

## Ethics

Animal experimentation: This study was performed in strict accordance with the recommendations in the Guide for the Care and Use of Laboratory Animals of the LANUV (Landesamt für Natur, Umwelt und Verbraucherschutz). Reference number: 84-02.04.2012.A192.

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
