## [Decision Letter]

Thank you for submitting your work entitled "A novel biosensor to study cAMP dynamics in cilia and flagella" for consideration by *eLife*. Your article has been favorably evaluated by Richard Aldrich (Senior editor) and three reviewers, one of whom, (David Clapham) is a member of our Board of Reviewing Editors.

The following individuals involved in review of your submission have agreed to reveal their identity: Aldeberan Hofer and Dejian Ren (peer reviewers).

The reviewers have discussed the reviews with one another and the Reviewing Editor has drafted this decision to help you prepare a revised submission.

Summary:

*Mloti*K1 channel that binds cAMP with K_d_~ 70 nM and cGMP with K_d_~500 nM. They generate transgenic mice expressing it in sperm flagellae and primary cilia. The results suggest that free cAMP in the flagellum is distinct from that in midpiece or head, and is dominated by soluble bicarbonate-stimulated adenylyl cyclase (AC10). This is a useful tool for monitoring free cAMP in sperm and cilia. This work is exciting because it represents a completely new type of cAMP sensor that doesn't rely on sequences from mammalian PKA, Epac or cyclic nucleotide-gated channels. An important feature of mlCNBD-FRET is its high affinity for cAMP. Null-point calibration estimated that resting cytosolic cAMP is around 35nM. This is perhaps the first accurate measurement of resting free cAMP, although this may reflect a mixture of resting cGMP and cAMP. Another useful feature is that the sensor protein can be isolated and used *in vitro* to monitor cAMP. The experiments in sperm cell and cilia nicely demonstrated the feasibility of using the probe *in vivo*, but these studies are preliminary and yielded less well-supported new information. All reviewers felt the paper was most appropriate as a 'Tools and Resource' paper.

1) The authors should extend the range of changes in K_d_ for cAMP binding to pH 8, since this can be achieved in alkaline environments in sperm flagella. Also the results with 25mM NaHCO_3_ are ambiguous. Figure 2 indicates a strange pH sensitivity of the reporter, and it's not certain how pH changes upon NaHCO_3_ addition. A control experiment to measure pH upon NaHCO_3_ treatment, and an independent test of FRET changes of mlCNBD-FRET using a maneuver known to change pH in a defined manner (e.g. NH3 or lactate) would be helpful. Does the cAMP-insensitive mlCNBD-FRET-R307Q respond to NaHCO_3_? One reviewer noted that in Figure 7, in the experiments where [Ca^2+^]_o_ is manipulated, the authors interpret the changes of FRET signals as changes in [cAMP]. It is possible that the reduction/increase in [Ca^2+^]_o_ also changes intracellular pH or [Ca^2+^], which in turn affects the FRET signal independent of changes in [cAMP] (Figure 2). A simple "negative" control is perhaps to repeat the experiment in the presence of NKH477. The inhibitor presumably should also block the FRET changes induced by increases in [Ca^2+^]_o_.

2) The data presented indicate that mlCNBD-FRET has quite high affinity for cAMP (66nM purified protein, 73nM in situ in cells) and is saturated at around 3µM cAMP. Of note, the probe also has relatively high affinity for cGMP (504nM), which is similar to some other reporters designed expressly for this purpose. The high affinity for cAMP is very useful for assessing subtle variations in resting cAMP, but is buffering of cAMP a problem? Although Prm1-mICNBD-FRET males were fertile, was flagellar beating altered by the expression of mlCNBD-FRET?

3) Figure 3 demonstrates that the biosensor works well when expressed in HEK cells. However, only maneuvers expected to saturate the sensor are shown. Since low affinity probes based on Epac are also saturated by these treatments that produce massive amounts of cAMP (probably 50- 100µM), it is not surprising that mICNBD-FRET also becomes saturated. It would be nice to see the response to physiological stimulation with sub-maximal agonist concentrations, and also the reversibility of the probe (not by permeabilization with digitonin, but through PDE activity). Figure 3 shows that the cAMP-insensitive mlCNBD-FRET-R307Q also demonstrates a modest decrease when digitonin is applied, calling into question whether the decreased baseline shown for the parent sensor reflects a true decrease in cAMP/cGMP below baseline.

4) The FLIM data are weak, with cerulean lifetime changes only from 2.44+/-02 to 2.38 +/-04 ns after large change in free cAMP. We suggest stating the FLIM data, but pointing out that it is not very suited to FLIM, unless the authors are able to obtain much more data supporting this claim. Thus that section of the figure can be removed.

---

## [Author Response]

1) The authors should extend the range of changes in K_d_ for cAMP binding to pH 8, since this can be achieved in alkaline environments in sperm flagella.

We have performed additional cAMP binding studies at pH 8 to determine the K_d_. The K_d_ at pH 8 is 50 ± 13 nM, indicating that the affinity for cAMP increases upon alkalization. Of note, it has been reported for other CNBDs that the affinity for cAMP depends on the pH (Gordon et al., 1996; Kaupp & Seifert, 2002). The information has also been included in the manuscript and the text has been changed accordingly.

Also the results with 25mM NaHCO_3_ are ambiguous. Figure 2 indicates a strange pH sensitivity of the reporter, and it's not certain how pH changes upon NaHCO_3_ addition. A control experiment to measure pH upon NaHCO_3_ treatment, and an independent test of FRET changes of mlCNBD-FRET using a maneuver known to change pH in a defined manner (e.g. NH3 or lactate) would be helpful. Does the cAMP-insensitive mlCNBD-FRET-R307Q respond to NaHCO_3_?

To test whether NaHCO_3_ evokes a change in the intracellular pH_i_, we measured changes in pH_i_ using BCECF in wild-type mouse sperm after stimulation with NaHCO_3_ and after stimulation with NH_4_Cl as a control. Stimulation with NH_4_Cl evoked a pronounced alkalization and, in turn, decreased the FRET ratio (Figure 5—figure supplement 1). The kinetics of both changes were similar, indicating that the change in FRET is evoked by the change in pH_i_. In contrast, stimulation with NaHCO_3_ only evoked a miniscule and slow change in pH_i_ (Figure 5—figure supplement 1) that was not comparable to the rapid changes in FRET ratio after bicarbonate stimulation (see Figure 6). These results support the notion that the changes in FRET evoked by NaHCO_3_ reflect changes in cAMP rather than pH_i_.

One reviewer noted that in Figure 7, in the experiments where [Ca^2+^]_o_ is manipulated, the authors interpret the changes of FRET signals as changes in [cAMP]. It is possible that the reduction/increase in [Ca^2+^]_o_ also changes intracellular pH or [Ca^2+^], which in turn affects the FRET signal independent of changes in [cAMP] (Figure 2). A simple "negative" control is perhaps to repeat the experiment in the presence of NKH477. The inhibitor presumably should also block the FRET changes induced by increases in [Ca^2+^]_o_.

To test whether [Ca^2+^]_o_ affects the intracellular pH, we measured pH_i_ using BCECF in wild-type mouse sperm after increasing or decreasing [Ca^2+^]_o_. As a control, sperm were stimulated with NH_4_Cl, which evoked a pronounced alkalization and, in turn, decreased the FRET ratio (Figure 5—figure supplement 1). Reducing [Ca^2+^]_o_ to 443 nM by addition of 3 mM BAPTA did not change the intracellular pH (Figure 7).

We are sorry, but we do not understand the suggestion using NKH477. NKH477 is an activator (not inhibitor) of tmACs that are absent in sperm flagella; NKH477 has no effect on sperm cAMP levels (see Figure 6).

2) The data presented indicate that mlCNBD-FRET has quite high affinity for cAMP (66nM purified protein, 73nM in situ in cells) and is saturated at around 3µM cAMP. Of note, the probe also has relatively high affinity for cGMP (504nM), which is similar to some other reporters designed expressly for this purpose. The high affinity for cAMP is very useful for assessing subtle variations in resting cAMP, but is buffering of cAMP a problem? Although Prm1-mICNBD-FRET males were fertile, was flagellar beating altered by the expression of mlCNBD-FRET?

This argument is well taken. As the reviewer also pointed out, Prm1-mlCNBD-FRET males are fertile, indicating that sperm function is not severely altered by the sensor. The flagellar beat frequency of wild-type and transgenic sperm was similar: wild-type: 8.4 ± 2.1 Hz, n = 8, 58 cells total; mlCNBD -FRET: 7.3 ± 3.0 Hz, n = 4, 40 cells total. The information has been added to [Supplementary-material SD1-data].

*3) Figure 3 demonstrates that the biosensor works well when expressed in HEK cells. However, only maneuvers expected to saturate the sensor are shown. Since low affinity probes based on Epac are also saturated by these treatments that produce massive amounts of cAMP (probably 50- 100µM), it is not surprising that mICNBD-FRET also becomes saturated. It would be nice to see the response to physiological stimulation with sub-maximal agonist concentrations, and also the reversibility of the probe (not by permeabilization with digitonin, but through PDE activity).*

We agree. Accordingly, we performed a dose-response relationship for stimulation with NKH477 to activate the tmACs. The EC50 for NKH477 is 3.6 ± 0.6 µM (n = 4). This data set has been included in Figure 3—figure supplement 1.

To show the reversibility of the sensor, we have alternatingly stimulated cells with isoproterenol followed by a wash-out with buffer. After wash-out, the FRET ratio was lowered and increased again after stimulation with isoproterenol, demonstrating the reversibility of the sensor. This data set has also been included in Figure 3—figure supplement 1.

Figure 3 shows that the cAMP-insensitive mlCNBD-FRET-R307Q also demonstrates a modest decrease when digitonin is applied, calling into question whether the decreased baseline shown for the parent sensor reflects a true decrease in cAMP/cGMP below baseline.

The mlCNBD-FRET-R307Q mutant has been generated as a control to visualize any unspecific effects on the FRET ratio that are independent of cAMP. The small decrease in baseline for the mlCNBD-FRET-R307Q mutant is probably due to a permeabilization artefact. However, Figure 3 shows a larger decline of the FRET ratio after permeabilization with digitonin from the plateau that is reached after stimulation with NKH477.

4) The FLIM data are weak, with cerulean lifetime changes only from 2.44+/-02 to 2.38 +/-04 ns after large change in free cAMP. We suggest stating the FLIM data, but pointing out that it is not very suited to FLIM, unless the authors are able to obtain much more data supporting this claim. Thus that section of the figure can be removed.

Although the changes in cerulean lifetime in the FLIM measurements are relatively small, they are nonetheless significant. To the best of our knowledge, such FLIM measurements with cAMP FRET sensors have not yet been reported. Thus, we prefer to leave this figure as it is. However, we have rephrased the text to: “In summary, mlCNBD-FRET is an exquisitely sensitive biosensor for measuring cAMP dynamics in the nanomolar range, preferably using fluorescence intensity techniques, but also using lifetime-based techniques.”